# Identification by High-Throughput Real-Time PCR of 30 Major Circulating *Listeria monocytogenes* Clonal Complexes in Europe

Benjamin Félix,[a] Karine Capitaine,[a] Sandrine Te,[a] Arnaud Felten,[b] Guillaume Gillot,[c] Carole Feurer,[d] Tijs van den Bosch,[e] Marina Torresi,[f] Zsuzsanna Sréterné Lancz,[g] Sabine Delannoy,[h] Thomas Brauge,[i] Graziella Midelet,[i] Jean-Charles Leblanc,[a] Sophie Roussel[a]

[a]ANSES, European Union Reference Laboratory for Listeria monocytogenes, Laboratory for Food Safety, Salmonella and Listeria Unit, University of Paris-Est, Maisons-Alfort, France
[b]ANSES, Ploufragan/Plouzané/Niort Laboratory, Viral Genetics and Bio-Security Unit, Université Européenne de Bretagne, Ploufragan, France
[c]ADRIA Food Technology Institute, Quimper, France
[d]IFIP–The French Pig and Pork Institute, Department of Fresh and Processed Meat, Le Rheu, France
[e]Wageningen Food Safety Research, Department of Bacteriology, Molecular Technology and Antimicrobial Resistance, Wageningen, The Netherlands
[f]National Reference Laboratory for Listeria monocytogenes, Istituto Zooprofilattico Sperimentale dell'Abruzzo e Molise "G. Caporale" Via Campo Boario, Teramo, Italy
[g]Microbiological National Reference Laboratory, National Food Chain Safety Office, Food Chain Safety Laboratory Directorate, Budapest, Hungary
[h]ANSES, Laboratory for Food Safety, IdentyPath Platform, Maisons-Alfort, France
[i]ANSES, Laboratory for Food Safety, Bacteriology and Parasitology of Fishery and Aquaculture Products Unit, Boulogne-sur-Mer, France

**ABSTRACT** *Listeria monocytogenes* is a ubiquitous bacterium that causes a foodborne illness, listeriosis. Most strains can be classified into major clonal complexes (CCs) that account for the majority of outbreaks and sporadic cases in Europe. In addition to the 20 CCs known to account for the majority of human and animal clinical cases, 10 CCs are frequently reported in food production, thereby posing a serious challenge for the agrifood industry. Therefore, there is a need for a rapid and reliable method to identify these 30 major CCs. The high-throughput real-time PCR assay presented here provides accurate identification of these 30 CCs and eight genetic subdivisions within four CCs, splitting each CC into two distinct subpopulations, along with the molecular serogroup of a strain. Based on the BioMark high-throughput real-time PCR system, our assay analyzes 46 strains against 40 real-time PCR arrays in a single experiment. This European study (i) designed the assay from a broad panel of 3,342 *L. monocytogenes* genomes, (ii) tested its sensitivity and specificity on 597 sequenced strains collected from 24 European countries, and (iii) evaluated its performance in the typing of 526 strains collected during surveillance activities. The assay was then optimized for conventional multiplex real-time PCR for easy implementation in food laboratories. It has already been used for outbreak investigations. It represents a key tool for assisting food laboratories to establish strain relatedness with human clinical strains during outbreak investigations and for helping food business operators by improving their microbiological management plans.

**IMPORTANCE** Multilocus sequence typing (MLST) is the reference method for *Listeria monocytogenes* typing but is expensive and takes time to perform, from 3 to 5 days for laboratories that outsource sequencing. Thirty major MLST clonal complexes (CCs) are circulating in the food chain and are currently identifiable only by sequencing. Therefore, there is a need for a rapid and reliable method to identify these CCs. The method presented here enables the rapid identification, by real-time PCR, of 30 CCs and eight genetic subdivisions within four CCs, splitting each CC into two distinct subpopulations. The assay was then optimized on different conventional multiplex real-time PCR systems for easy implementation in food laboratories. The two assays will be used for frontline identification of *L. monocytogenes* isolates prior to whole-genome sequencing. Such assays are of great interest for all food industry stakeholders and public agencies for tracking *L. monocytogenes* food contamination.

Address correspondence to Benjamin Félix, benjamin.felix@anses.fr.

The authors declare no conflict of interest.

**KEYWORDS** *Listeria monocytogenes*, MLST, food, food processing environment, high-throughput real-time PCR, clonal complexes, conventional multiplex real-time PCR

*L*isteria monocytogenes is a ubiquitous bacterium that can be found in many habitats (1, 2). This Gram-positive facultative intracellular bacterium is responsible for listeriosis, a serious foodborne zoonosis affecting both humans and animals (3). The pathogen is transmissible to humans through the consumption of contaminated food (3). Food contamination can originate from either raw plant or animal matter or from food processing environments (FPEs) (4). The ability of *L. monocytogenes* to grow at low temperatures, form biofilms, and persist in food processing plants makes this bacterium a significant challenge for food safety (5, 6). Many food sectors have been hit hard by a series of *L. monocytogenes*-related food poisoning outbreaks in Europe in recent years (7–10). In 2019, listeriosis was the most serious zoonosis with the highest case fatality rate (13%) among outbreak-related illnesses (11).

*L. monocytogenes* is a genetically heterogeneous species divided into 13 serotypes and four phylogenetic lineages, of which lineages I and II are the most frequently encountered (12). Multilocus sequence typing (MLST) classifies *L. monocytogenes* into clonal complexes (CCs) and sequence types (STs), which are systematically used to describe its population structure (13). STs are defined as the unique association of alleles from seven housekeeping genes, and a CC is described as a cluster of STs sharing at least six alleles (14). CCs descend from a common ancestor and have accumulated differences predominantly through mutations (14–17). CCs evolve slowly over large temporal and geographic scales (15, 16). Hypervirulent and hypovirulent CCs have been identified by combining epidemiological, clinical, and experimental approaches (18, 19). CC1, CC2, CC4, and CC6 are among the known hypervirulent CCs most likely to cause disease, particularly central nervous system or maternal-neonatal listeriosis (19). In contrast, hypovirulent CCs, such as CC9 and CC121, merely cause disease in highly immunocompromised patients and show limited virulence in humanized mouse models (19). CC nomenclature has been harmonized internationally (14), and data can be directly compared during epidemiological surveillance and outbreak investigations (7–9). CC classification provides crucial information on strain virulence (18, 19) and on the possible origin of contamination (20).

In Europe, in addition to the 20 CCs known to account for the majority of outbreaks and sporadic cases in humans (21) and animals (22–25), 10 CCs are frequently reported in the food chain from the primary production stage to the final products (19, 20, 26–29). These 30 CCs (11 of lineage I and 19 of lineage II) cover the four risk food sectors (meat, fish, milk and cheese, and fruits and vegetables) underlined by the European Food Safety Authority (11). They can persist in FPEs in various food sectors, thereby potentially posing a serious challenge for the agrifood industry (20, 30–44).

MLST is the reference method for the identification of *L. monocytogenes* CCs, but its major drawbacks are the cost and time to carry it out, from 3 to 5 days for laboratories that outsource sequencing. A conventional multiplex PCR assay has been developed as an alternative to MLST (45) but can differentiate only 11 of the 30 major CCs circulating in the food chain. Therefore, there is a need for a rapid and reliable method to identify these 30 major CCs. Such a method would be of great interest for all food industry stakeholders and public agencies for tracking *L. monocytogenes* contamination throughout the food chain.

Real-time PCR technology offers the possibility to rapidly detect *L. monocytogenes* with higher specificity, sensitivity, and reliability than conventional PCR using agarose gel-based detection (46). A plethora of real-time PCR assays has been developed for detecting and quantifying *L. monocytogenes* in various food matrixes (46–48) and in water and environmental surfaces (49). These methods enable the identification of species or the five major molecular serogroups (50, 51). Some methods have been developed for screening *L. monocytogenes* strains related to an outbreak, as applied recently in Italy (52). However, to date, no assay based on real-time PCR has been available for rapid identification of CCs.

In this study, a rapid and reliable high-throughput real-time PCR assay was developed for the identification of the 30 major *L. monocytogenes* CCs encountered in FPEs

and final food products in the European Union. In a single experiment, 46 strains can be simultaneously tested for the identification of the CCs and the molecular serotype, as well as the confirmation of the species. This assay provides results in less than 1 day starting from a bacterial culture. To meet the needs of the food industry, we optimized the assay on conventional multiplex real-time PCR systems for easy implementation in routine diagnostic laboratories. This study describes in detail the sensitivity, specificity, and validation of both assays (high-throughput and conventional real-time PCR assays) by following international standards EN ISO 16140 (53) and EN ISO 17025 (54). Finally, this study meets the requirements laid out in the Minimum Information for Publication of Quantitative Real-Time PCR Experiments (MIQE) guidelines (55).

## RESULTS

**Development of the high-throughput real-time PCR assay. (i) Design of primers and probes.** Thirty-four sets of primers and probes were specifically designed for this study (Table 1). A set of primers and probes was selected for each CC except for four (CC14, CC1, CC37, and CC121) for which two sets were designed to discriminate between mutational subdivisions within each CC (named CC-SD) (Table 1). Sequences as well as amplicon size and targeted genes are listed in Table 1. Possible cross-reactions that could not be avoided were identified *in silico* for 49 STs (Table 2).

**(ii) LoD.** The 95% limit of detection (LoD$_{95}$) was determined for all the real-time PCRs and ranged between 30,000 and 300,000 copies/$\mu$L, using a cycle threshold (C$_T$) of $\leq$25. The minimal DNA concentration recommended for the method was thus set to 30,000 copies/$\mu$L, corresponding to 0.1 ng/$\mu$L of *L. monocytogenes* genomic DNA. The minimal DNA concentration of 0.1 ng/$\mu$L was compatible with the amount of DNA produced by the extraction methods used in this study. The upper limit for the LoD was set to 300,000 copies/$\mu$L (i.e., 1 ng/$\mu$L of *L. monocytogenes* genomic DNA) to avoid possible contamination during the BioMark microfluidic chip loading process.

**(iii) Analytical sensitivity.** The real-time PCR results were 100% sensitive, on strain panel C (SP-C) (Table 2), for all but three CCs, namely, CC9 (93%), CC193 (91%), and CC204 (85%).

For CC9, of the 24 ST9 strains analyzed (Table 2), two false-negative results were observed in strains 15SEL871LM and 21SEL229LM (SP-C; see Table S1 in the supplemental material) isolated in France from meat products. The genomic locus targeted by the real-time PCR assay to identify CC9 was absent in both strains due to a 3.5-kb deletion in strain 21SEL229LM and a 4.4-kb deletion in strain 15SEL871LM. These deletions were observed in 2.8% (4/142) of the ST9 strain genomes available in GP-A and -B. They were isolated in France, Italy, the Netherlands, and the USA from fish and meat products. The strain genomes were otherwise genetically close to the other ST9 genomes ($<$74 allelic differences [AD]) available in GP-A and -B.

For CC193, one single ST662 strain generated a false-negative result (strain 01EB168LM isolated in France from smoked salmon) (SP-C; Table S1). The CC193 primer and probe locus was located over a *guaA* insertion site. For strain 01EB168LM, the PCR was blocked by a 10-kb integrative element inserted in *guaA*. In addition to 01CEB168LM, two ST662 and one ST796 strain genomes (GP-B) displayed the same integrative element. These strains were reported from Canada and France from fish products and from the USA from clinical cases, respectively. The ST662 and ST796 strains reported here were genetically distant from the other CC193 strains, with more than 1,000 AD.

For CC204, of the 13 ST204 strains analyzed, two false-negative results were observed for strains L00500 and L01157 isolated in the Netherlands in a fish product and a poultry meat product, respectively (SP-C; Table S1). The genomic locus targeted by the real-time PCR assay to identify CC204 strains was located in a 43-kb genomic integrative element inserted in the gene *inlK*. This integrative element was absent in both strains (L00500 and L01157) and in 4.7% (5/106) of the ST204 strain genomes available in GP-A and -B. These strains were reported from Australia, Austria, France, Germany, and the USA from meat and dairy products, when known. The genomes of these strains, for which the 43-kb

**TABLE 1** Primers and TaqMan probes used in the high-throughput real-time PCR conventional multiplex real-time PCR assays

| Target CC | Target gene | Name[c] | Sequence (5′–3′)[b] | $T_m{}^a$ (°C) | Amplicon length (bp) | Multiplex group |
|---|---|---|---|---|---|---|
| CC1 | AAA family ATPase (bacterial chromosome) | CC1_P<br>CC1_F<br>CC1_R | Cy5(FAM)-TTCCAGCACTCAATGCAATCGC-BBQ<br>GTTCGATAGTGTCATAGGA<br>GCTCTCTATTCAATATTGGTAA | 67.1<br>58.0<br>59.7 | 100 | IVb-1 |
| SD_CC1 | Alkylated DNA nucleotide flippase Atl1 (bacterial chromosome) | SD_CC1_P<br>SD_CC1_F<br>SD_CC1_R | FAM-AACATTCGAAGCGTAACCAAATTACG-BBQ<br>CAGACTCGTAGGTGCTAC<br>GACGTGTMCATTCTCTTTA | 67.9<br>61.3<br>58.0 | 154 | |
| CC2 | SIR2 superfamily (bacterial chromosome) | CC2_P<br>CC2_F<br>CC2_R | FAM-TCATCTTGTCCGATAGGTTCTGATTCT-BBQ<br>GCGTTTATTGGAAGGAAA<br>TGGGAAAGATTTCTTCTCA | 70.3<br>54.3<br>55.9 | 99 | IVb-1 |
| CC3 | Hypothetical protein (bacterial chromosome) | CC3_P<br>CC3_F<br>CC3_R | Cy5(FAM)-AGTCGCTTTGACGAATATCAAACTCAC-BBQ<br>ACCCAAATAGATCAAAGC<br>CGGATTCTCTCTATTCTTG | 70.3<br>54.3<br>58.0 | 119 | IIb-1 |
| CC4 | Cytosine-C5-specific DNA methylases (bacterial chromosome) | CC4_P<br>CC4_F<br>CC4_R | FAM-TGCCTCCTACCAACTGTACTGAAG-BBQ<br>CATCGTAGCCTTTCATC<br>GGAACTAACCGAGGATTA | 70.2<br>56.4<br>56.4 | 129 | IVb-2 |
| CC5 | AcrR family (bacterial chromosome) | CC5_P<br>CC5_F<br>CC5_R | HEX-AGACACATTAATTCCGCTTGGCAA-BBQ<br>CCTTGCTAGCTTCTGTAG<br>GAAGGTACTTTACAGACAAA | 67.5<br>58.8<br>58.4 | 99 | IIb-1 |
| CC6 | D12 class N6 adenine-specific DNA methyltransferase (bacterial chromosome) | CC6_P<br>CC6_F<br>CC6_R | HEX-AACGGATTCTATTAAACACGCAAGCAA-BBQ<br>GGCAGTGTTTGATACATG<br>CTGGTAGAATAGATTACTTTAGAC | 68.7<br>56.4<br>63.3 | 127 | IVb-1 |
| CC7 | Type I restriction-modification system, DNA methylase subunit (bacterial chromosome) | CC7_P<br>CC7_F<br>CC7_R | Cy5(FAM)-AACTGCAACTCCAGAGTCAACATAAT-BBQ<br>GGTGAAATATGAGTAAATGGA<br>GAACCTATATTTTGAGGCATTA | 67.9<br>58.4<br>59.7 | 138 | IIa-3 |
| CC8 | Hypothetical protein (bacterial chromosome) | CC8_P<br>CC8_F<br>CC8_R | Cy5(FAM)-AGTCACAGAAACTTCTAAGCCGG-BBQ<br>GGTACGGGTAGTTTGTTA<br>GCCTTTTCAATGAAGTGAA | 67.9<br>58.0<br>55.9 | 117 | IIa-1 |
| CC9 | Hypothetical protein (bacterial chromosome) | CC9_P<br>CC9_F<br>CC9_R | FAM-TCTTCTCCGAGTGTATACGCCT-BBQ<br>CAGGATTTAAGACCCCTAC<br>CTCTCTCAAATTGAATGCTTA | 67.9<br>60.0<br>58.4 | 100 | |
| CC11-ST451 | Trans-acting positive regulator Mga HTH (bacterial chromosome) | CC11-ST451_P<br>CC11-ST451_F<br>CC11-ST451_R | FAM-TTAAGACTCGGCGCATGTTGCTGTGCAC-BBQ<br>GATGGGAGTTAATGATTTTATGGATA<br>ACACCTATTCTTTCTTGATTATACAG | 74.8<br>65.1<br>65.1 | 167 | IIa-7 |
| CC14-ST14-206-399 | Glucosamine-6-phosphate deaminase (bacterial chromosome) | CC14-ST14-206-399_P<br>CC14-ST14-206-399_F<br>CC14-ST14-206-399_R | FAM-TCAGGACAAATCAGTGCATTTGGCC-BBQ<br>GATGCAACTGCTATTAGG<br>GTTCGTACATTCGCTTAG | 70.8<br>56.4<br>56.4 | 88 | IIa-3 |
| CC14-ST91-160-360 | ATP phosphoribosyltransferase regulatory subunit (bacterial chromosome) | CC14-ST91-160-360_P<br>CC14-ST91-160-360_F<br>CC14-ST91-160-360_R | HEX-TTTTCCACTTTAAGTTGCTCATTACC-BBQ<br>AGTGGTAAGGTTACATTA<br>CGAGCGAAATAAAATTAAATG | 67.1<br>51.9<br>56.7 | 132 | IIa-4 |
| CC18 | Hypothetical protein (bacterial chromosome) | CC18_P<br>CC18_F<br>CC18_R | Cy5(FAM)-TCTAACTCCTCTGTGTAAGAAGCAATT-BBQ<br>GTCACATTGGTTATATTTCAAG<br>CAGGTTTATTTCACTAGTTTG | 68.7<br>59.7<br>58.4 | 148 | IIa-4 |
| CC19-ST398 | Oligoendopeptidase, pepF/M3 family (bacterial chromosome) | CC19-ST398_P<br>CC19-ST398_F<br>CC19-ST398_R | HEX(FAM)-CGTTAGTTTATCAATACGTCGCAACTTTAGC-BBQ<br>CTTGCTTCCGCTGATATCAATG<br>GTCCCAAACACGGTCAGAAA | 73.4<br>65.1<br>63.4 | 99 | IIa-7 |

**TABLE 1** (Continued)

| Target CC | Target gene | Name[c] | Sequence (5′–3′)[b] | $T_m^a$ (°C) | Amplicon length (bp) | Multiplex group |
|---|---|---|---|---|---|---|
| CC20 | Hypothetical protein (bacterial chromosome) | CC20_P | Cy5(FAM)-TCTAGCCTGTTCAATTTCTTGATTGG-BBQ | 67.9 | 121 | IIa-5 |
| | | CC20_F | TGTCCTAATAGTGTAAGCA | 55.9 | | |
| | | CC20_R | CGTGAAAATGAACAACTAAA | 55.2 | | |
| CC21 | Hypothetical protein (bacterial chromosome) | CC21_P | FAM-TCAACTTGCTTGTTTTAAACCAAGA-BBQ | 65.1 | 111 | IIa-5 |
| | | CC21_F | GCAACTAAAATAACTATCTCAA | 57.7 | | |
| | | CC21_R | GGATGAAAATTACTGATGAAG | 58.4 | | |
| CC26 | SIR2 superfamily (bacterial chromosome) | CC26_P | Cy5(FAM)-AGACATAATGAATCATGGACGCTTCTT-BBQ | 68.7 | 131 | IIa-6 |
| | | CC26_F | ACACGACGTATGACTTTA | 54.3 | | |
| | | CC26_R | CAGCATCTTCAAACAGAG | 56.4 | | |
| CC29 | AdoMet S-adenosylmethionine-dependent methyltransferases (bacterial chromosome) | CC29_P | FAM-TTGACGCTGAACTTGCTAATGC-BBQ | 65.1 | 114 | IIa-6 |
| | | CC29_F | AACGGCTATTAAACGGAG | 56.4 | | |
| | | CC29_R | GGCAAAGTTACTACAGTTG | 58.0 | | |
| CC31 | Hypothetical protein (bacterial chromosome) | CC31_P | Cy5(FAM)-ATGAAACGAGCTAAATCTCCTCAATT-BBQ | 66.7 | 148 | IIa-2 |
| | | CC31_F | GAGTGTATGGCATATGAAAG | 59.3 | | |
| | | CC31_R | GATCGTTGATAGAGAATTAAAATC | 61.6 | | |
| CC37 | Ribonuclease YeeF family protein (bacterial chromosome) | CC37_P | FAM-TCAGGCAGCACTTCATAATCCA-BBQ | 65.1 | 114 | IIa-2 |
| | | CC37_F | CCAGAGAATGGCTAGATA | 56.4 | | |
| | | CC37_R | GAACCAATAGAAGAATTGATAC | 59.7 | | |
| SD_CC37 | Intergenic space (bacterial chromosome) | SD_CC37_P | FAM-ACTGCAAATGATTCCGATGGAATTTAC-BBQ | 68.7 | 154 | |
| | | SD_CC37_F | GCGAGTGAACTAGTGAAAA | 58.0 | | |
| | | SD_CC37_R | GAAGCTGCTTCAGTAGAAG | 60.0 | | |
| CC54 | Cof subfamily of IIB subfamily of haloacid dehalogenase (bacterial chromosome) | CC54_P | HEX-AGCCTCCCGTACCGTAAACCGGT-BBQ | 73.3 | 102 | IVb-2 |
| | | CC54_F | AGGACATATTAGATGTTCGTTCTG | 65.3 | | |
| | | CC54_R | GCTTCACCAACACTTAGCATA | 62.5 | | |
| CC59 | Nickel ABC transporter substrate-binding protein (bacterial chromosome) | CC59_P | Cy5(FAM)-AAAGAATCCCGACGAAACGCT-BBQ | 65.1 | 123 | IIb-2 |
| | | CC59_F | CAGCAAAAGACAGCAGATA | 58.0 | | |
| | | CC59_R | AGCCAGAATAAATAAATTTACTTAC | 60.9 | | |
| CC77 | InlH/InlC2 family class 1 internalin (bacterial chromosome) | CC77_P | FAM-ACAGAACCAATTCCTCCAACCAA-BBQ | 65.9 | 141 | IIb-2 |
| | | CC77_F | CACGAATCAAACTGTGAA | 54.3 | | |
| | | CC77_R | CTTCGCAGGCATTTTATC | 56.4 | | |
| CC87 | Rolling-circle replication initiation protein (bacterial chromosome) | CC87_P | FAM-ATCCTTTGAGTGATAAACATCGCCTAC-BBQ | 70.3 | 124 | IIb-1 |
| | | CC87_F | GTGACACCATGTAAATCTC | 58.0 | | |
| | | CC87_R | GCAGAAAACTTGGAATGA | 54.3 | | |
| CC101 | RNA polymerase subunit (bacterial chromosome) | CC101_P | HEX-CACTCTTAATGTTATGTGCTAAGCCG-BBQ | 69.6 | 100 | IIa-5 |
| | | CC101_F | ATGGCACTTGAATTATTCA | 53.9 | | |
| | | CC101_R | GCCATACGTTCGATTTTAC | 58.0 | | |
| CC121 | AlwI restriction endonuclease (bacterial chromosome) | CC121_P | FAM(HEX)-TTAGATTGCTACTACCGCCAATT-BBQ | 64.2 | 146 | IIa-1 |
| | | CC121_F | ATGGCTACTGAATATATCCC | 59.3 | | |
| | | CC121_R | TCGGAATTTATCATTATATGTTCTA | 60.9 | | |
| SD_CC121 | Hypothetical protein (bacterial chromosome) | SD_CC121_P | FAM-TTTGACATGAATCGAAATCACTTCA-BBQ | 64.2 | 98 | |
| | | SD_CC121_F | TCACCAAGACAAGTTTTAG | 55.9 | | |
| | | SD_CC121_R | CGATGATATCGCTTGAAAA | 55.9 | | |
| CC155 | Peptidase S8 (bacterial chromosome) | CC155_P | HEX-ATATTCAGAATCCATCCCTATTTGCG-BBQ | 67.9 | 133 | IIa-1 |
| | | CC155_F | GTCAGAGTCGAATTCATTA | 55.9 | | |
| | | CC155_R | TCTGGAATTTTCAAAAGTATTG | 57.7 | | |

**TABLE 1** (Continued)

| Target CC | Target gene | Name^c | Sequence (5′–3′)^b | $T_m$^a (°C) | Amplicon length (bp) | Multiplex group |
|---|---|---|---|---|---|---|
| CC193 | Archaea DNA helicase HerA (bacterial chromosome) | CC193_P | HEX-TGATGAGGAACCATATCATTTCCAATG-BBQ | 68.7 | 133 | IIa-6 |
| | | CC193_F | CTGTCATGTGTTATCCTTG | 58.0 | | |
| | | CC193_R | TGGGAATAACGAGTCAATA | 55.9 | | |
| CC199 | Putative AbiEii toxin (bacterial chromosome) | CC199_P | Cy5(FAM)-AGACTCTCCACTTCCAGCAAACGCTTCTGT-BBQ | 77.1 | 193 | IIa-7 |
| | | CC199_F | CGGAGCATTCACTATATCATTTACA | 65.9 | | |
| | | CC199_R | GTCAGTTGGATGTTAGACCAAA | 63.4 | | |
| CC204 | Hypothetical protein (bacterial chromosome) | CC204_P | HEX-TGTGGACAACTTCTCTAATTTCATCT-BBQ | 67.1 | 101 | IIa-3 |
| | | CC204_F | CCTCTTGGTACTTCTAAATTATC | 62.6 | | |
| | | CC204_R | CAGAGCCGAAGATTATCC | 58.8 | | |
| CC224 | HATPase super family (bacterial chromosome) | CC224_P | HEX-TCTTGTCCAAATTGTTTCACTATTATCGTAAGTA-BBQ | 71.7 | 106 | IIb-2 |
| | | CC224_F | GAACGTATCTCTCAGTAGC | 61.3 | | |
| | | CC224_R | GAAGGATTTATTAGAAATGAAAGTA | 60.9 | | |

^a The melting temperature ($T_m$) was calculated as described by Kibbe et al. (101) (http://biotools.nubic.northwestern.edu/OligoCalc.html).

^b Dye and quencher reported in the table are indicative and used for the conventional multiplex real-time PCR assay. The high-throughput real-time PCR assay used FAM and HEX reporting dye. Dye changes between the two methods were reported under bracket.

^c _P, probe; _F, forward primer; _R, reverse primer.

**TABLE 2** Analytical specificity and sensitivity results of the high-throughput real-time PCR assay for 597 *L. monocytogenes* strains (SP-C)

| MLST CC and subdivision (no. of strains) | Analytical specificity | Analytical sensitivity | Cross-reaction with analytical confirmation (no. of strains) [*in silico*-predicted cross-reaction][b] |
|---|---|---|---|
| CC1 SD_1 (10) | 1 | 1 | |
| CC1 SD_2 (9) | 0.992 | 1 | CC183 (2), ST213 (1), ST773 (1) [CC373[a], ST1125] |
| CC2 (17) | 1 | 1 | |
| CC3 (11) | 0.998 | 1 | CC1000 (1) [CC489, ST558, ST1046, ST1041, CC1211] |
| CC4 (15) | 1 | 1 | |
| CC5 (14) | 1 | 1 | |
| CC6 (15) | 1 | 1 | |
| CC7 (26) | 1 | 1 | [CC373[a]] |
| CC8 (31) | 0.998 | 1 | ST1110 (1) |
| CC9 (28) | 0.998 | 0.93 | ST184 (1) [ST395, ST1331[a]] |
| CC11-ST451 (19) | 1 | 1 | |
| CC14-ST14-206-399 (16) | 0.998 | 1 | CC689[a] (1) [ST843] |
| CC14-ST91-160-360 (12) | 1 | 1 | |
| CC18 (13) | 1 | 1 | |
| CC19-ST398 (7) | 1 | 1 | [CC1127] |
| CC20 (14) | 0.995 | 1 | ST19 (1), ST173 (3) [ST226[a], ST364, ST378, ST1021, ST1071, ST1078] |
| CC21 (13) | 0.989 | 1 | CC403 (5) |
| CC26 (16) | 1 | 1 | [ST376, ST790, CC912, ST1024, ST1331[a]] |
| CC29 (14) | 1 | 1 | [CC344, ST1082] |
| CC31 (13) | 1 | 1 | |
| CC37 SD_1 (11) | 1 | 1 | |
| CC37 SD_2 (9) | 0.992 | 1 | CC321 (5) [ST648, ST828, ST1068] |
| CC54 (12) | 1 | 1 | |
| CC59 (11) | 1 | 1 | |
| CC77 (10) | 1 | 1 | |
| CC87 (7) | 0.998 | 1 | CC88 (1) |
| CC101 (13) | 1 | 1 | [CC90, ST671, ST1127] |
| CC121 SD_1 (13) | 1 | 1 | |
| CC121 SD_2 (15) | 0.998 | 1 | CC689[a] (1) |
| CC155 (22) | 1 | 1 | |
| CC193 (11) | 0.996 | 0.91 | CC124 (2) [ST798] |
| CC199 (9) | 1 | 1 | [CC739, ST1331] |
| CC204 (13) | 1 | 0.85 | [ST798] |
| CC224 (10) | 0.997 | 1 | ST581 (1), ST585 (1) [ST226[a], ST1118] |
| Non-targeted *L. monocytogenes* and non-*L. monocytogenes* (90) | | | |
| Total | Strain panel C (597) | | |

[a]ST or CC identified as a cross-reaction by two distinct primer and probe sets.

[b]A number within parentheses after an CC or ST represents the number of strains tested by primer and probe sets. STs and CCs within brackets were predicted *in silico* as a cross-reaction but not tested on a strain.

genomic integrative element is deleted, were genetically close to the other CC204 strain genomes (<36 AD) in GP-A and -B.

**(iv) Analytical specificity.** The specificity was 100% for 22 of the 34 primer and probe sets (Table 2). For the 12 other sets, the specificity ranged from 98.9 to 99.8% (CC1 SD_2, CC3, CC8, CC9, CC14-ST14-206-399, CC20, CC21, CC37 SD_2, CC87, CC121 SD_2, CC193, and CC224) (Table 2). Of the 49 possible cross-reactions observed *in silico*, 16 were confirmed by analyzing 27 corresponding strains (Table 2, SP-C; Table S1). For the others, it was not possible to confirm them, because the strains were not available (Table 2). For the 62 closely related strains in SP-C, with 7 AD or less, the results obtained by the high-throughput real-time PCR assay were similar to the MLST data.

**(v) Performance of the assay.** Variations in the extraction methods did not affect the assay's performance. For the 25 strains extracted by the three different methods, all primer and probe sets were effective in amplifying their respective targets.

In total, 526 SP-D.1 unsequenced strains were tested using high-throughput real-time PCR (Table 3). Of the 289 strains isolated from food products and FPEs, the CC was identified for 272 (94%). Of the 237 strains isolated from ruminants, the CC was identified for 182 (78%). The remaining strains, for which the CC was not identified by high-throughput real-time PCR (Table 3), were sequenced, leading to the confirmation that they did not belong

**TABLE 3** Performance results of the high-throughput real-time PCR assay and of the conventional multiplex real-time PCR assay for the panels of *L. monocytogenes* involving 526 strains (SP-D.1) and 77 strains (SP-D.2), respectively[a]

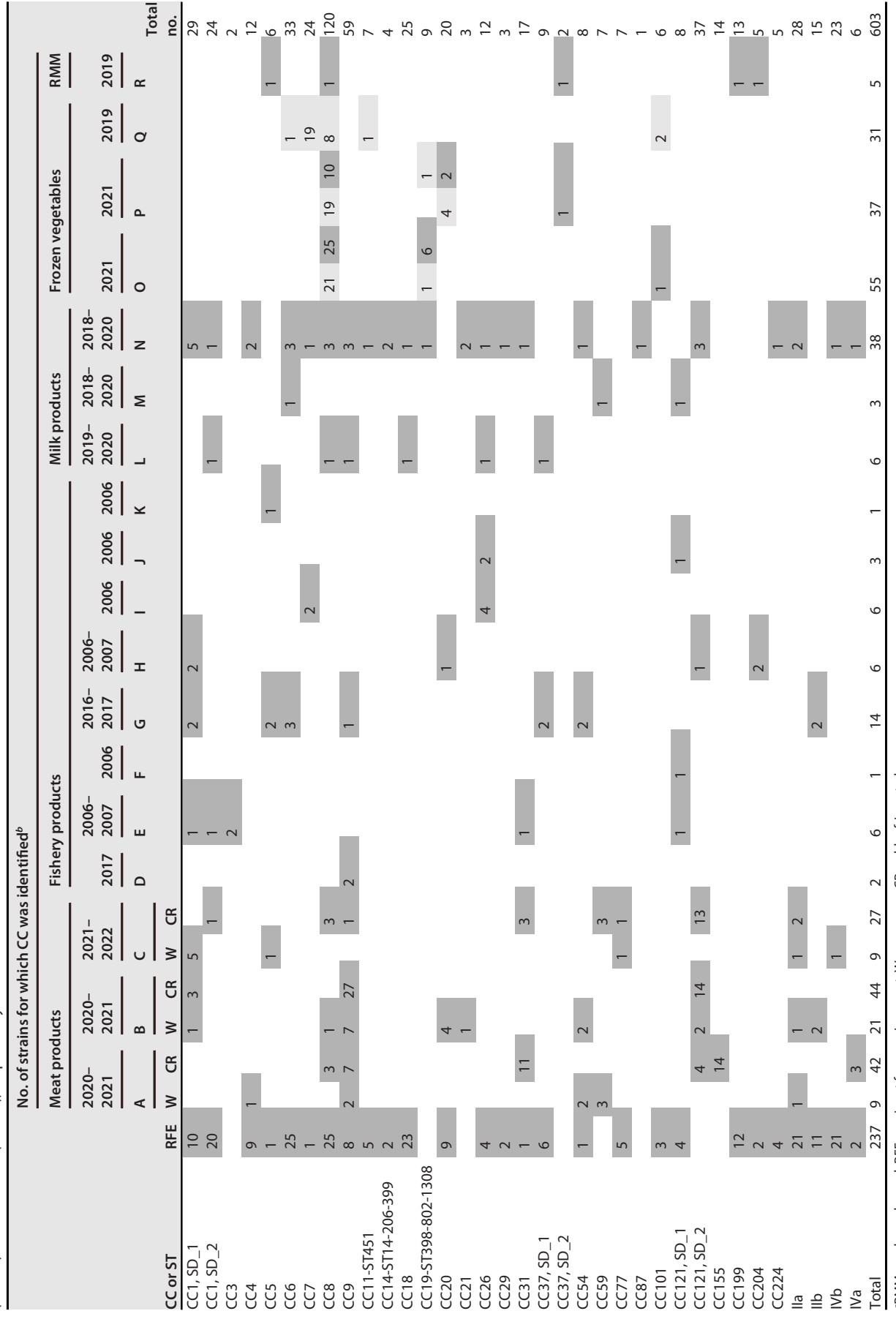

[a]RMM, ready-made meal; RFE, ruminant farm environment; W, warm area; CR, cold refrigerated area.

[b]Results are number of strains identified for the indicated food sector, period of sampling, processing plant identifier (A to R), and area of sampling (W or CR). Dark-shaded values indicate analysis performed by high-throughput real-time PCR assay (SP-D.1, *n* = 526); light-shaded values indicate analysis performed by multiplex conventional real-time PCR assay (SP-D.2, *n* = 77).

to the 30 targeted CCs. They belonged to CC11-ST11, ST191, CC207, CC217, CC379, CC415, CC517, and CC991.

For the strains isolated from processing plants, the data obtained showed a heterogeneous distribution of the CCs between the different sampling areas in the same plant and between plants (Table 3). The 14 CC155 strains were isolated only from processing plant A, most likely indicating the persistence of these strains associated with the cold room across two different years in the plant. Among the 18 processing plants analyzed, nine CC19-ST398 strains were isolated from processing plants N, O, and P (Table 3). Processing plant N showed a large diversity of contamination, including 18 of the 30 identified CCs, whereas in processing plants O and P, the CC8, CC19-ST398, and CC20 strains were the major contaminants (Table 3). CC1, CC9, CC26, CC31, CC54, and CC121 were observed in at least five different processing plants or a ruminant farm environment (Table 3). Of the six CC1 strains isolated from processing plant C, one was CC1-SD_2 and the others were CC1-SD_1. The SD_2 strain was isolated in the cold processing room and the others were isolated in the temperate part of the plant, indicating that two different CC1 populations had colonized these areas (Table 3).

**Development of the multiplex conventional real-time PCR assay. (i) Limit of detection.** Using the same parameters of the high-throughput assay, in particular the $C_T$ of ≤25, the LoD$_{95}$ values obtained with the multiplex conventional real-time PCR assay were greater than 30,000 copies/$\mu$L (>0.1 ng/$\mu$L of genomic DNA). These results were not consistent with the yield of the DNA extraction method, in particular the cell lysis extraction, which can produce DNA up to a concentration of 0.1 ng/$\mu$L. To solve this issue, the $C_T$ was set at ≤30. Using this lower $C_T$ value, a LoD$_{95}$ between 30,000 and 300,000 copies/$\mu$L was established for all multiplex real-time PCR assays.

**(ii) Identification of CC using the conventional multiplex real-time PCR assay on a LightCycler 480 instrument II and TaqMan 7500 fast system.** From the 597 sequenced strains (SP-C), 92 (Table S1) were analyzed on a LightCycler 480 instrument II to compare the results obtained with the MLST data. All primer and probe sets were effective in amplifying their respective targets. For the 77 unsequenced strains (Table 3; SP-D.2), the CC was identified for all strains. For the 373 sequenced strains (SP-E) analyzed on a TaqMan 7500 fast system (Table S1), all primer and probe sets successfully amplified their respective targets.

**(iii) Use of the multiplex conventional real-time PCR assay by national reference laboratories in an investigative context.** Exported frozen corn was reported as contaminated by *L. monocytogenes* in Hungary in 2021 (56). This strain was sequenced and classified as CC19-ST398 by the Hungarian National Reference Laboratory (HU NRL; the Hungarian National Food Chain Safety Office, NEBIH). An official investigation was carried out by the Hungarian authority at three frozen vegetable processing plants, O, P, and Q (Table 3). From these plants, the HU NRL isolated 123 strains and analyzed 77 of the strains using the conventional real-time PCR assay (Table 3). The remaining strains were analyzed by the European Union Reference Laboratory (EURL) for *L. monocytogenes* by the high-throughput real-time PCR assay. Among the 123 strains, 8 were positive with the CC19-ST398 primer and probe set (Table 3). These strains were subsequently sequenced, and their genomes were compared with that of the strain initially isolated from the exported frozen vegetables. These analyses identified only processing plant O as the origin of the contamination, with strains sharing less than 3 AD with the strain initially isolated from exported frozen vegetables. The strain isolated from processing plant P differed by 32 AD.

Several human clinical cases caused by a strain of ST173 were reported over a multiyear period in the Netherlands, and two fish product producers were suspected of being the source of the contaminations. Subsequently, in 2022, the Dutch Food Safety Authority (NVWA) performed official audits in these two fish processing plants. Environmental samples were taken by swabbing a total of 25 high-risk locations and objects, such as industrial crates, trays, drains, conveyor belts, and cutting machines. After sampling, the swabs were analyzed using a classical microbiological method

equivalent to ISO 11290-1, which resulted in seven *L. monocytogenes*-positive samples. Five colonies were isolated from each *L. monocytogenes*-positive sample. The 35 strains isolated from these two processing plants in the Netherlands were analyzed by the associated Netherlands (NL) NRL. The CC20 primer and probe set also identifies ST173 strains as a cross-reaction (Table 2). Therefore, this set was used to identify the ST173 isolates of the outbreak. Of the 35 strains analyzed, 6 were positive for CC20, and the other strains yielded only the molecular serotype, which was IIa. The CC20-positive strains were analyzed by whole-genome sequencing (WGS), and their relatedness with the outbreak was confirmed. These analyses clearly identified one of the processing plants as the origin of the contamination.

## DISCUSSION

The two real-time PCR assays developed for this study proved to be able to (i) identify the 30 major circulating *L. monocytogenes* CCs in the European Union (EU) agrifood industry, five molecular serotypes, and four CC subdivisions, (ii) confirm the species *L. monocytogenes*, (iii) provide a fast and reliable method that generates results in less than 1 day starting from a bacterial culture, and (iv) meet the needs of the food industry with an optimized PCR method that can be easily implemented in routine diagnostic laboratories.

The 30 CCs also cover more than 77% of the environmental and animal isolates reported in Europe (13) and between 80 and 70% of strains isolated from ruminant pathological cases in Slovenia, Latvia, Switzerland, and the United Kingdom (24, 25, 28). The 30 CCs targeted in this study also cover more than 80% of the food isolates reported in large multicontinental studies (16, 17), more than 70% of the isolates reported in the USA from food and FPEs (31, 57), and 89% of the isolates reported in China (58) from meat products. However, the 30-CC set covers 30% of isolates contaminating rivers on the West Coast of the USA (59). It showed that in some cases the assay should be expanded to include additional CCs of interest.

Thirty-four new primer and probe sets were specifically designed, facilitated by the increased number of bacterial genomes available in public repositories. The design was based on a worldwide extended collection of 3,342 genomes by using two original bioinformatics tools developed in-house based on k-mer (see "Identification of favorable k-mer positions" in Materials and Methods) or core genome MLST (cgMLST) allele selection (see "Identification of favorable point mutations" in Materials and Methods). This panel was selected to be international and to cover the very large genomic diversity reported in food strains. Therefore, the assays developed in this study were designed to be able to identify CCs in food strains circulating worldwide.

Of the 34 primer and probe sets, 28 were designed to target unique sequences on the *L. monocytogenes* chromosome, in accordance with the approach applied by Doumith et al. (60) for the method of molecular serotyping primer design. This method, considered as the reference method worldwide, has proven to be reliable and robust, particularly in several European proficiency testing trials performed over many successive years (61). The other sets targeting CC14-ST14, CC14-ST91, CC101, and the PCR designed for "subdivisions" of CC1, CC37, and CC121 were located on several single-nucleotide-polymorphism loci and not on unique sequences. In contrast to the other primers and probes placed on unique sequences, these PCRs showed some background amplification (noise) and interpretation was based on internal and positive controls.

The assays reported five strains giving a false-negative result for three primer and probe sets: CC9, CC204, and CC193. For CC9 and CC204, four strains giving a false-negative result were observed, for ST9 (*n* = 2) and ST204 (*n* = 2). The CC9 primer and probe set was placed on the gene *lmo1118*, formerly identified by Doumith et al. (62) as a characteristic of IIc strains (51, 60). In a former study conducted in our laboratory (63, 64), of 135 IIc strains isolated in France from foods between 2005 and 2006, 3 were also negative for the primer and probe characteristic of IIc strains. Our hypothesis is that these strains display genetic changes similar to those of the CC9 false-negative

strains. Similar CC9 false-negative strains were reported by Chenal-Francisque et al. in 2015 (45). For CC204, no similar genetic profiles have been described in the literature. For CC193, one strain giving a false-negative result was also found. These strains belonged to ST662 and ST796 and are genetically distant (>1,000 AD) from the other CC193 strains, the majority of which belong to ST193. The ST662 and ST796 strains were associated with CC193 in the nomenclature, as they share 6 alleles in common but in fact are genetically distant from the other CC193 strains and should not be assigned to this CC. For ST662, one false-negative strain was confirmed analytically by high-throughput real-time PCR. For ST796, one false-negative strain was observed only *in silico*, not by PCR.

All false-negative results obtained for CC9, CC193, and CC204 were related to the insertion or deletion of integrative elements in their respective primer and probe loci. Such integrative elements have often been described in the *L. monocytogenes* genome (65–68). For these three primer and probe sets, no alternative designed to avoid false-negative results was available. The only alternative was to complement the PCR with a second one, targeting a genetic region characteristic of the false-negative strains. However, given the very limited number of false-negative strains observed by our method, our general approach and design of the primer and probe sets were deemed robust. The possible false-negative strains will be described to alert users to alert the users. Compared with *Salmonella* (69, 70), *L. monocytogenes* is less affected by frequent recombination events (71, 72) and shows remarkable consistency between the molecular serotyping methods devised by Doumith et al. (60) and the genomic markers, making molecular serotyping concordant with MLST and MLST alternatives (14). This genomic consistency is reflected in the very few discrepancies observed. In this study, 49 STs were associated with cross-reactions. Of the 49 STs, 27 were confirmed by high-throughput real-time PCR and the others were found only *in silico* without PCR confirmation. As shown in the example of the ST173 outbreak strain investigation in the Netherlands, these cross-reactions can be used to identify STs or CCs not primarily targeted by our assay.

WGS of *L. monocytogenes* has become a very powerful tool for national surveillance, outbreak detection, or tracking down of the listeriosis sources (73–75). However, WGS remains expensive for routine use (consumables cost >€100 for each isolated strain, excluding the cost of analysis) and time-consuming (approximately 5 days from receipt of strain to final results) and requires specific skills as well as substantial computer storage space. Moreover, the level of precision obtained by WGS is not always required with respect to the problems encountered in the food industry. In comparison with WGS, the high-throughput real-time PCR assay developed in this paper is more cost-effective (approximately €4 per strain for consumables), less time-consuming (results obtained in less than 1 day starting from a bacterial culture), and less labor-intensive (2,304 reactions per BioMark chip) and has only limited computational requirements. The conventional multiplex real-time PCR assay can be carried out in a single day, is suitable for routine analysis, and is cost-effective (between €2 and €8 for consumables). For surveillance laboratories that collect large numbers of food strains, these assays allow them to screen the strains to be sequenced based on the CC obtained, thus minimizing the volume of sequencing experiments, saving time and resources. However, in comparison to WGS, these assays provide identification of only 30 CCs, while 256 CCs and 1,037 single STs were identified in the MLST online database (https://bigsdb.pasteur.fr) in February 2023. Nevertheless, the 30 CCs identified are those most found worldwide (>80%) in food products, animal carriage, and the environment.

The two real-time PCR assays were developed as part of the activities of the EURL Laboratory for *L. monocytogenes* (https://eurl-listeria.anses.fr) in close collaboration with its French and EU NRLs in charge of *L. monocytogenes* surveillance in specific food sectors. As shown in the investigation of product contamination conducted in Hungary, the results were both produced and compared by the EURL for *L. monocytogenes* and the HU NRL by high-throughput and conventional real-time PCR assays, respectively. This comparison

illustrated the complementarity of the two methods, facilitating efficient cooperation between partners.

Here, we demonstrated that the conventional real-time PCR assay was compatible with different thermocyclers, including the fast PCR system (55 min instead of 90 min for an entire cycle). Two NRLs (NL and Italy [IT]) participated in the development of this assay on strains isolated in their respective countries. Both laboratories now use this test routinely. Following training and support actions organized by the EURL, five NRLs have been already trained but have not yet implemented the method as routine (the Cyprus State General Laboratory, Greek Ministry of Rural Development and Food Directorate of Veterinary Center of Athens, the Hungarian National Food Chain Safety Office, the Northern Macedonia Veterinary Faculty Food Laboratory, and the Kosovo Food and Veterinary Laboratory).

Recently developed portable fast thermocyclers are particularly useful for performing rapid real-time PCR analysis in the field (76–78). We developed the multiplex real-time PCR assay to be compatible with such mobile fast PCR thermocyclers. This optimization may enable CC identification directly, at the processing plant, linked with the contaminated stage, for instance, in a slaughterhouse or in a farm. Moreover, with such PCR systems, the multiplex real-time PCR assay can be adopted by more and more laboratories. It is thus planned to further minimize the cost of the assay by reducing the reaction master mix volume from 20 $\mu$L to 15 $\mu$L.

The two assays developed in this paper were validated for the analysis of genomic DNA purified from an isolated strain. We plan to evaluate and optimize the assays for the analysis of DNA extract from contaminated food matrix, environmental samples, or animal samples without the strain isolation step. To date, *L. monocytogenes* detection methods in complex samples (food product, environmental, and animal samples) can confirm only the *L. monocytogenes* species (79, 80). The standard detection method EN ISO 11290-1:2017 requires 72 h to be performed (81), while methods based on real-time PCR require only 48 h (82). The optimization of our assays for complex samples will offer the possibility of identifying the CC in 48 h, with or without an enrichment step, and analyses are under way. This optimized method may be able to be used to investigate situations of multiple *L. monocytogenes* contamination, in samples and enrichment broths. Furthermore, direct analysis of complex samples may be of great help to better understand the hotly debated issue of interstrain competition in enrichment broths (83, 84). Application to matrices also opens the possibility for direct quantification of *L. monocytogenes*. This approach depends on DNA extraction efficiency and possible coextraction of PCR inhibitors.

**Conclusions.** Our fast, accurate, and valuable assays constitute a further step toward a better understanding and management of the health risks associated with *L. monocytogenes* for surveillance and contamination control in the agrifood industry. The methods do not cover the described whole MLST diversity within *L. monocytogenes* species but enables the typing of the 30 most abundant CCs found worldwide in food products. The wide use of these methods should contribute to (i) defining the worldwide distribution of CCs along the food chain, (ii) providing an accurate view of *L. monocytogenes* population structure in food, and (iii) anticipating the emergence of new genetic types. These assays represent key tools for assisting surveillance laboratories in the field in (i) differentiating food strains representing the most significant health risks, (ii) understanding the entry and transfer of *L. monocytogenes* in the food chain, (iii) assessing the risks represented by the strains detected, (iv) tracing the origin of contamination during outbreak investigations, and (v) adapting microbiological and hygiene management plans in processing plants and then selecting the most appropriate control measures accordingly.

## MATERIALS AND METHODS

**Genomic DNA extraction.** The DNA of 25 strains from 25 distinct CCs, 21 from targeted CCs and 4 from nontargeted CCs (see Table S1 in the supplemental material), were simultaneously extracted using three different extraction methods for comparison purposes. For the three extraction methods, the DNA

concentration was determined by fluorometric measurement using a Life Technologies Qubit 3 fluorimeter and a DNA high-sensitivity kit (Thermo Fisher Scientific, Saint-Herblain, France).

**(i) Cell lysis genomic DNA extraction method.** The InstaGene matrix (Bio-Rad, Marnes-la-Coquette, France) kit, the version for bacterial DNA extraction, was used. The manufacturer's recommendations were to start with three *L. monocytogenes* colonies on nonselective agar medium cultured for 24 h. The double-stranded DNA extraction yield ranged from 0.1 to 10 ng/$\mu$L.

**(ii) Isopropanol/ethanol genomic DNA extraction method.** The Wizard SV genomic DNA purification system extraction kit (Promega, Charbonnières-les-Bains, France), the version for Gram-positive bacterial DNA extraction, was used. The manufacturer's recommendations were to start with 1.4 mL of *L. monocytogenes* culture of pure strain in brain heart infusion (BHI) grown overnight, with an optical density at 600 nm between 1.0 and 1.8. An additional prelysis step was applied using a solution of lysozyme (Roche, Meylan, France) (2 mg/mL) and EDTA (40 mM) for 60 min at 37°C. The double-stranded DNA extraction yield ranged from 10 to 600 ng/$\mu$L.

**(iii) Silica membrane genomic DNA extraction method.** The Qiagen DNeasy Blood & Tissue extraction kit (Qiagen, Les Ulis, France), the version for Gram-positive bacterial DNA extraction, was used. The manufacturer's recommendations were to start with 1 mL of *L. monocytogenes* culture of pure strain in BHI broth overnight. An additional prelysis step was applied using a solution of Tris-HCl (20 mM), sodium EDTA (2 mM), Triton X-100 (1.2%), and lysozyme (0.2 g/L) for 30 min at 37°C. The double-stranded DNA extraction yield ranged from 10 to 400 ng/$\mu$L.

**Development of the high-throughput real-time PCR assay. (i) Design of primers and probes.** *(a) Identification of favorable k-mer positions.* The genomic panel (GP) included 954 genomes (GP-A) of *L. monocytogenes* strains isolated from human cases, food, animals, and the environment. This panel was constituted to cover both the genomic diversity within the 30 targeted CCs (Table 1) and that in a wide geographical area in Europe and worldwide (Table S2). Fifty-seven percent of the genomes came from strains isolated in 18 European countries, and the rest came from 19 non-European countries. Of the 954 GP-A genomes, 598 were downloaded from the two public bacterial genome databases: the National Center for Biotechnology Information Sequence Read Archive (NCBI-SRA; https://www.ncbi.nlm.nih.gov/) and the European Molecular Biology Laboratory–European Bioinformatics Institute database (EMBL-EBI; https://www.ebi.ac.uk/ena). They were downloaded in fastq format on 6 December 2018 via the BioNumerics version 7.6.3 calculation engine (bioMérieux Applied Maths, Sint-Martens-Latem, Belgium) set up at the French agency for food, environmental and occupational health & safety (ANSES). The other 356 genomes were sequenced by ANSES as part of surveillance, monitoring, outbreak investigations, and collaborative research projects. Genome assembly was carried out with BioNumerics version 7.6.3 using the ANSES calculation engine. The calculation engine was run with SPADES v.3.7.1 (85). Only genomes between 2.8 and 3.1 Mb, consistent with the *L. monocytogenes* genomic size, were used. For each CC, at least 15 genomes were selected (Table S2). These genomes were selected to maximize ST diversity and geographical spread within each CC.

The 954 GP-A genomes were processed using an ANSES in-house-developed bioinformatics pipeline based on python script (not published). Briefly, all the genome assemblies were fragmented into 300-bp fragments (called k-mers). K-mers specific and sensitive to each CC were identified by the clustering tool cdhit (86) using an 80% similarity threshold. The k-mer was present in all the strains of a certain CC but absent in all the strains of the other CCs. A single k-mer was chosen for each CC without preselection criteria.

*(b) Identification of favorable point mutations.* When no k-mer could be selected, another bioinformatics pipeline was used to design primers and probes. Whole-genome MLST (wgMLST) (using a 4,807-gene scheme) was applied to the 954-genome panel (GP-A), using BioNumerics version 7.6.3 and the ANSES calculation engine. Allele specific and sensitive of the targeted CC were selected using character type tools of BioNumerics 7.6.3 and then aligned using the software sequence alignment functionality. Based on the alignment, primers and probes were placed on the allelic discriminant mutation. Primer and probe parameters were designed and validated as described below in "*In silico* validation and production."

*(c) In silico validation and production.* TaqMan hydrolysis probes and primers were designed on the selected k-mer, using Beacon designer 8.2 software (Premier Biosoft, San Francisco, USA), with hairpins, self-dimers, and multiplex dimers set to 6, 10, and 8 kcal/mol, respectively. Amplicon secondary structure was avoided, with the setting "avoid template structure" parameter of the software. The usual recommendations were followed for primer and probe design as described by Poitras and Houde (87). Primer and probe sets were validated *in silico* using the Basic Local Alignment Search Tool (BLAST; NCBI toolkit version 2.2.28) (88), in BioNumerics 7.6.3, using a genomic panel of 2,388 genomes (GP-B). They were directly uploaded from the NCBI RefSeq assembly database on 28 June 2019 and included 2,299 *L. monocytogenes* (286 different STs and 89 CCs) and 89 non-*L. monocytogenes* strain genomes. Only primers and probe sets without *in silico* false-negative cross-detection were conserved. False-positive cross-detections were accepted if limited to rare CCs. If the primer and probe set was not satisfactory at this stage, another k-mer was selected for a new design and subsequent BLAST validation. Probes were labeled with either reporting dye 6-carboxyfluorescein (FAM; maximum absorbance at 495 nm and maximum emission at 520 nm) or hexachlorofluorescein (HEX; maximum absorbance at 535 nm and maximum emission at 556 nm), with a BlackBerry Quencher covalently bound at the 3′ end. The primers and probes were supplied by TIB Molbio, Berlin, Germany (https://www.tib-molbiol.de).

**(ii) Real-time PCR conditions.** The analyses were carried out on a high-throughput microfluidic real-time PCR system, the BioMark HD (Fluidigm, San Francisco, CA, USA). The system ran 48:48 chips. A 6-$\mu$L PCR mix containing 4.5 $\mu$M primers, 2 $\mu$M probes, and the 2× assay loading reagent (Fluidigm PN 85000736) was prepared for each set of primers and probes. A 6-$\mu$L sample mix containing 3 $\mu$L of 2× PerfeCTa qPCR ToughMix Low Rox (https://www.quantabio.com; Quantabio, Beverly, MA, USA), 0.3 $\mu$L of sample loading reagent (Fluidigm PN 85000746), and 2.7 $\mu$L of diluted DNA (0.1 to 1 ng/$\mu$L) was

prepared per sample. Then, 5 $\mu$L of the PCR mix and 5 $\mu$L of the sample mix were transferred to the chip inlets and loaded with the Integrated Fluidic Circuits (IFC) controller. After loading, the chip was transferred to the BioMark instrument. The PCR run started with 10 min at 95°C, followed by 40 cycles at 95°C for 15 s and 60°C for 1 min. The data from the BioMark instrument were analyzed with the Fluidigm real-time PCR analysis software (Fluidigm) by using manually defined thresholds set up at 0.005 of the normalized reporting value ($\Delta$RN). A real-time PCR was considered positive when the $C_T$ was less than or equal to 25.

Each PCR run included positive controls consisting of six pairs of probes and primers previously published for classifying the *L. monocytogenes* strains into the five major molecular serogroups and for confirming the genus *Listeria* and the species *L. monocytogenes* (51). Each PCR run also included a negative control (DNA free) and internal controls using a pBluescript II SK plasmid mixture carrying the PCR genetic amplicons flanked by 50-bp segments on both sides. Cloning was carried out by the GeneCust, Boynes, France (https://www.genecust.com/).

**(iii) LoD.** The LoD was established for each real-time PCR. The material used was prepared in a dilution series using positive-control plasmid solution, with dilutions ranging from 0.3 to 300,000 copies/$\mu$L ($10^{-6}$ to 1 ng/$\mu$L genomic DNA concentration equivalence). The number of genome copies was estimated from the DNA quantity through fluorometric measurement, considering the genome size of *L. monocytogenes* equal to 2.94 million nucleotides and the molecular weight of one nucleotide equal to 660 g/mol (71, 89). The same methodology was applied for plasmid copy number estimation. Each LoD test was repeated 20 times for each real-time PCR. The LoD$_{95}$ was calculated according to the ISO 16140 LODPOD tool (90). The concentration range which gave the best reaction efficiency and linearity was used to test and validate the assay.

**(iv) Sensitivity and specificity.** The analytical sensitivity and specificity were determined following PCR veterinary diagnostic validation standard NF U47-600 (91). A strain panel (SP) of 597 strains (SP-C) was used. It included 587 strains collected and was sequenced in 24 European countries (Table S1) by 27 partners, including food institutes and National Reference Laboratories (NRLs).

Of the 597 strains (SP-C), 480 strains were selected to represent the genomic diversity of *L. monocytogenes* observed in Europe, in food products, animals, and the environment, among the 30 targeted CCs. They covered 76 STs (Table S1). The strains had been isolated over a period of more than 55 years (i.e., 1964 to 2021) across all stages of food production, from primary production to the final products. Of the 480 strains, 62 were chosen for their low genetic distance—the lowest possible, i.e., <7 AD—based on the cgMLST scheme defined by Moura et al. (73). The 7-AD limit was used as the reference under which two strains are considered epidemiologically related (73). The remaining strains showed genetic distances greater than 7 AD.

Of the 597 strains (SP-C), the other 117 strains remaining included 65 *L. monocytogenes* strains with CCs different from the 30 targeted CCs covering 43 STs (Table S1). They also included 27 non-*L. monocytogenes* (92–97) and 25 non-*Listeria* strains (98, 99) (Table S1). The non-*L. monocytogenes* and non-*Listeria* strains were selected because they were known to be frequent food chain contaminants and can potentially be isolated along with *L. monocytogenes* (98–100).

**(v) Performance.** The performance of the assay was calculated using a panel of 526 strains (SP-D.1) that have not previously been sequenced (Table 3). Most had been isolated between 2016 and 2022. In total, 289 strains were isolated from meat, fish, milk, vegetables, and ready-made food-processing plants located in France and Hungary and 237 from ruminant primary production in France.

**Development of the conventional multiplex real-time PCR assay.** The conventional multiplex real-time PCR assay consisted of 3 duplex and 10 triplex PCRs. The multiplex associations between the primer and probe sets are listed in Table 1. Primer and probe sequences were the same as those used in the high-throughput assay. An additional dye, cyanine 5 (Cy5; maximum absorbance at 646 nm and maximum emission at 669 nm), was added for multiplexing purposes (Table 1). The interactions between multiplexed primer and probe sets were verified using Beacon Designer 8.2 software, with the same settings applied for simplex PCR. The positive and negative controls were those used in the high-throughput real-time PCR assay. The assay was designed to be performed in several steps according to the investigation context (Fig. 1). The assay was tested on two different thermocyclers, the LightCycler 480 instrument II (Roche Diagnostics, Meylan, France) and the TaqMan 7500 fast system (Thermo Fisher Scientific, Villebon-sur-Yvette, France).

**(i) LightCycler 480 instrument II.** The reaction mixture was prepared by mixing 0.3 $\mu$L of each probe (20 $\mu$M) and primer (20 $\mu$M) with 10 $\mu$L of master mix (10×) (PerfeCTa qPCR ToughMix Low Rox) completed with molecular biology-grade water up to 18 $\mu$L per well and 2 $\mu$L of DNA at a concentration between 0.1 and 1 ng/$\mu$L. The thermal amplification program was strictly identical to that of the high-throughput assay. Given the limited risk of contamination in multiplex PCR, the $C_T$ threshold was adapted and set to 30.

The LoD was determined by following the same methodology as that used for the development of the high-throughput assay. The following parameters were adapted: the LoD was established with a dilution series ranging from 3 to 300,000 copies/$\mu$L ($10^{-5}$ to 1 ng/$\mu$L genomic DNA concentration equivalence). Each real-time PCR was repeated four times.

*(a) Validation of the assay.* A panel of 92 sequenced strains of the SP-C (Table S1) was used to validate the method.

*(b) Performance of the assay.* The performance of the assay was measured on 77 strains (SP-D.2) that had not previously been sequenced and were isolated from FPEs in Hungary (Table 3).

**(ii) TaqMan 7500 fast system.** To meet the needs of field laboratories and food industries, the conventional multiplex real-time PCR assay was optimized on a fast PCR system, providing a result in 55 min instead of 90 min, using a LightCycler 480 instrument II.

The reaction mixture was prepared with each probe and primers at 0.3 $\mu$M, GoTaq (Promega) master

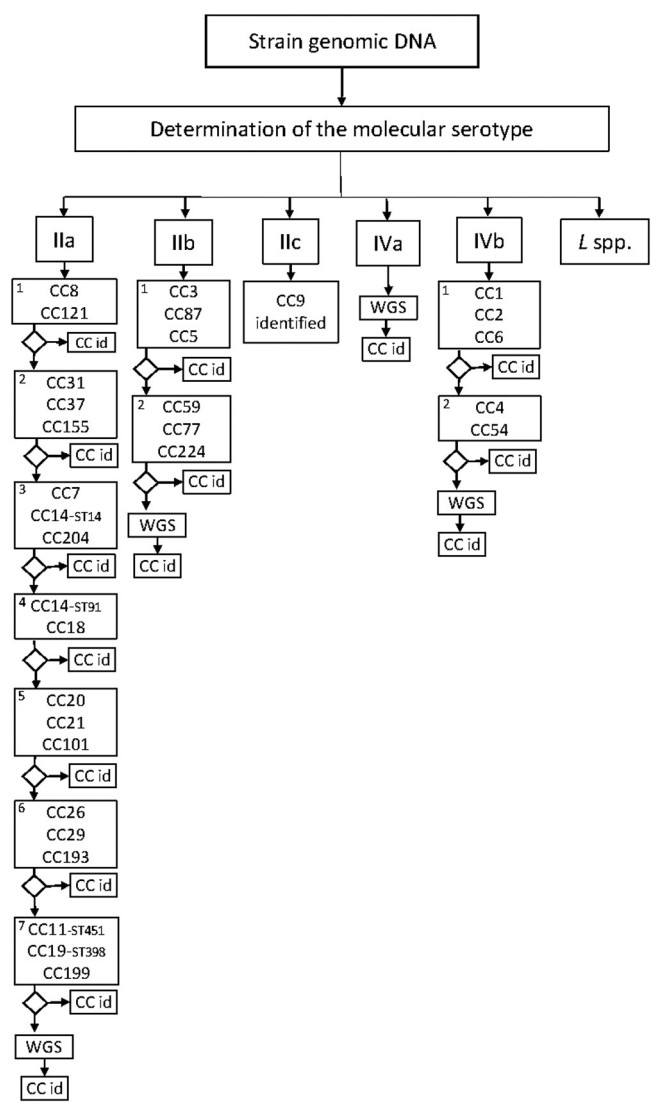

**FIG 1** Flow chart for the interpretation of the real-time multiplex PCR assay. The molecular serotype determined at this stage followed the PCR scheme of Vittulo et al. (51). CC numbers within each square represent the muliplex PCR. "CC id" means CC identified. The number in the upper left corner of each square refers to the multiplex number reported in Table 1.

mix (1×), 2 $\mu$L of DNA at a concentration between 0.1 and 1 ng/$\mu$L, in a 20-$\mu$L final volume. The fast PCR thermal amplification program included a 2-min activation step at 95°C, 40 cycles at 3 s at 95°C and annealing/extension for 30 s at 60°C, for a global run time of 55 min, with a heating speed set at 3°C/s.

*(a) Validation of the assay.* A panel of 373 previously sequenced strains (SP-E) was used for method validation (Table S1). The analyses were performed using the same experimental settings at three different laboratories: at ANSES, which is the European Union Reference Laboratory for *Listeria monocytogenes* (EURL for *L. monocytogenes*), the Italian NRL (IT NRL; the Istituto Zooprofilattico Sperimentale dell'Abruzzo e Molise "G. Caporale," IZSAM), and the Netherlands-associated NRL (NL NRL; the Wageningen Food Safety Research [WFSR]) (Table S1).

## SUPPLEMENTAL MATERIAL

Supplemental material is available online only.
**SUPPLEMENTAL FILE 1**, PDF file, 0.5 MB.
**SUPPLEMENTAL FILE 2**, PDF file, 0.7 MB.

## ACKNOWLEDGMENTS

We thank the staff of the European Union Reference Laboratory for *Listeria monocytogenes* for their contribution to the achievement of this project. We thank 11 NRLs for their advice during method development: the Belgian Institute for Health Sciensano (BE), the Agroscope

Institut (CH), the German Federal Institute for Risk Assessment (BfR) (DE), the Cyprus State General Laboratory (CY), the Istituto Zooprofilattico Sperimentale dell'Abruzzo e Molise "G. Caporale" (IT), the Wageningen Food Safety Research (associated NRL) (NL), National Institute for Public Health and the Environment (NL), the Portuguese National Institute of Agricultural and Veterinary Research (PT), the Veterinary Faculty of Ljubljana (SI), the Swedish Food Agency (SE), and the State Veterinary and Food Institute (SK). We thank the Austrian Agency for Health and Food Safety (AGES, AT NRL), the Estonian Veterinary and Food Laboratory (VetLab, EE NRL), the Finnish Food Authority, Laboratory and Research Division, Microbiology Unit (FI NRL), le Laboratoire National de Santé (LNS, LU NRL), the Institute of Food Safety, Animal Health and Environment (BIOR, LV NRL), the Norwegian Veterinary Institute (NVI, NO NRL), the Veterinary Faculty of Skopje (MK NRL), and the Flanders Research Institute for Agriculture, Fisheries and Food (ILVO) for providing strains. We acknowledge Guylaine Leleu from the ANSES Laboratory for Food Safety, Boulogne-sur-Mer site, for isolation of strains from fish processing plants and DNA extraction; Pierre-Emmanuel Douarre, Karol Romero, and Mai-Lan Tran from the ANSES Laboratory for Food Safety, Maisons-Alfort site, for their technical help for the support in genomic analysis, the strain DNA extractions, and the high-throughput real-time PCR analysis, respectively; Nathalie te Loeke and Kimberley van Kessel for the analysis and data treatment performed on the strains isolated in the Netherlands; Nassim Mouhali from the ADRIA Dévelopment laboratory for the extraction of strains from processing plants, as well as Pauline Robin from the French Pig and Pork Institute for her help in strain DNA extractions and multiplex PCR tests; Bertrand Lombard from the ANSES Strategy and Programs Department, Research and Reference Division, for his advice on the implementation of the ISO 16140 standard; and Katleen Vranckx from Applied Maths for her contribution to selecting the genome panels from NCBI-SRA and EMBL-ENA.

This work was conducted as part of the annual work program of the European Union Reference Laboratory (EURL) for *Listeria monocytogenes* (period 2020 to 2022).This work was cofunded by the European Union. The EURL for *L. monocytogenes* funded the biological material exchange and analysis performed jointly with other NRLs for multiplex real-time PCR validation. The validation study was partially funded by the Confederation of Marine Fisheries and Aquaculture Product Processing Industries (CITPPM). It funded the validation study performed on high-throughput real-time PCR. Wageningen Food Safety Research was funded by the Netherlands Food and Consumer Product Safety Authority. It funded part of the analysis performed in the Netherlands for multiplex PCR method validation and application.

The views and opinions expressed in this article are those of the authors only and do not necessarily reflect those of the European Union or the European Health and Digital Executive Agency (HaDEA). Neither the European Union nor HaDEA can be held responsible for them.

All authors read and approved the final manuscript. Benjamin Félix coordinated the project, performed the *in silico* analysis and primer and probe design, trained the partner laboratories, constructed the validation protocol according to the ISO 16140 standard, and wrote and edited the manuscript. Karine Capitaine performed most of the technical analyses, including high-throughput and multiplex real-time PCR analysis and data treatment. Sandrine Te participated in developing the validation protocol according to the ISO 16140 standard. Arnaud Felten performed the k-mer analysis. Guillaume Gillot, Carole Feurer, Tijs van den Bosch, Marina Torresi, Zsuzsanna Sréterné Lancz and Graziella Bourdin provided strain DNA from processing plants and contributed to project design. Sabine Delannoy provided technical advice for primer and probe design and high-throughput real-time PCR analysis. Jean-Charles Leblanc contributed to project design and revised the manuscript. Sophie Roussel contributed to the project design and the writing and editing of the manuscript.

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
