## [Reviewer comments · Microbiology Spectrum]

Microbiology Spectrum

Identification by high-throughput real-time PCR of 30 major circulating *Listeria monocytogenes* clonal complexes in Europe

Benjamin Félix, Karine Capitaine, Sandrine Te, Arnaud Felten, Guillaume Gillot, Carole Feurer, Tijs van den Bosch, Marina Torresi, Zsuzsanna Sréterné Lancz, Sabine Delannoy, Thomas Brauge, Graziella Midelet, Jean-Charles Leblanc, and Sophie ROUSSEL

Corresponding Author(s): Benjamin Félix, Anses

Review Timeline:

Submission Date:	November 3, 2022
Editorial Decision:	January 24, 2023
Revision Received:	February 21, 2023
Editorial Decision:	March 27, 2023
Revision Received:	March 31, 2023
Accepted:	April 14, 2023

Editor: Adelumola Oladeinde

Reviewer(s): The reviewers have opted to remain anonymous.

Transaction Report:

DOI: <https://doi.org/10.1128/spectrum.03954-22>

January 24, 2023

Mx. Benjamin Félix
Anses
Maisons-Alfort Laboratory for food safety, Salmonella and Listeria Unit, University of Paris-Est
14 rue pierre et marie Curie
Maisons-alfort 94701
France

Re: Spectrum03954-22 (Identification by high-throughput real-time PCR of 30 major circulating *Listeria monocytogenes* clonal complexes in Europe)

Dear Mx. Benjamin Félix:

Thank you for submitting your manuscript to Microbiology Spectrum. I have received feedback from two experts in the field. The two reviewers agreed that the research presented is relevant to the field and that the conclusions of the study were supported by sufficient data. Nonetheless, there are several comments by the reviewers and editor that must be addressed. In addition, the authors should ensure that the manuscript is formatted following ASM's guideline. Currently, I do not think ASM's formatting guideline was adhered to.

Link Not Available

Sincerely,

Adelumola Oladeinde

Journals Department
Reviewer comments:

Reviewer #1 (Comments for the Author):

Please see below the suggestions / comments.

Line-49: please put the hyphen between "real time" to make it uniform with other similar entries

Lines-50-82-83: please put "s" to CC

Line-121: please add more specifications on the 25 strains extracted

Line-144: in the sentence missing the verb, please insert it

Line-157: please check the correct wording of the sentence

Lines 190-192: In these sentences you talk about 48 samples tested, while in the abstract and in line 110, you refer to 46 samples, check and possibly specify

Lines-230 and 436: standardize the way in which references are reported (year with or without brackets)

Line-233: please put "s" to CC

Line-251: please change "an" to "and"

Line-352: please change 300 000 to 300,000

Line-369: please insert the space between "the" and "77"

Line-435: please put the hyphen between "false negatives" to make it uniform with other similar entries

Line-441: please change 1000 to 1,000

Line-465: please put "s" to CC

Line-576: please put "s" to strain

Line-604: please consider updating the reference (8) with the correct version referring to 2020 version

Line-754: please consider updating the reference (51) with the correct version, Part 1 is referred only to Vocabulaire; moreover the link doesn't work

Line-896: please standardize the title character of Table 1

Conclusions: please consider to add the limitation of the study or of the assays performed.

Reviewer #2 (Comments for the Author):

This study describes two novel real-time PCR methods (high-throughput and multiplex PCRs), which can rapidly discriminate 30 clonal complexes that are frequently encountered in the food chain. The methods developed by the authors will definitely attract a great interest from the public health sector and food industry by allowing to identify the CC of a strain without relying on the whole genome sequencing. Also, primers and probes were systematically designed using a large panel of genomes and the novel methods were thoroughly validated in various aspects. However, the current manuscript could be further improved by addressing weaknesses shown below.

1. The conventional multiplex real-time PCR was not sufficiently described, at least not to the degree that a reader can replicate this method after reading this manuscript. For instance, "3 duplex and 10 triplex PCR reactions" in lines 243 and 244 were not specified in the text or tables and "several steps" comprising this real-time PCR (lines 249 and 250) were not adequately delineated. Also, Figure 1, which was cited in lines 249 and 250, did not clearly convey the process and thus needs to be revised. Given that this conventional real-time PCR method was developed so that diagnostic laboratories can perform this assay routinely, this could be a major pitfall that could reduce the usefulness of this study. In the Materials and Methods section in general, more details are needed. See specific comments.

2. If the authors have included the features/functions of the inserted or deleted genes when discussing false-negative strains, it could have pleased many readers, including this reviewer, who are interested in the genetic diversity of *L. monocytogenes* population.

3. Show the GitHub site for the in-house bioinformatics pipeline, if available.

4. Many statements, in particular in the Results and Discussion sections, should be worked on to improve clarity. See specific comments.

5. References are often missing. See specific comments.

6. Cite tables and figures more frequently in Results.

Specific comments:

Keywords

Line 25: Change "Clonal complexes" to "Clonal Complexes".

Abstract

Line 35: Change "serogroup of the strain" to "serogroup of a strain".

Line 39: Change "its sensitivity and its specificity" to "its sensitivity and specificity".

Lines 46-47 and 56-57: Re-write "food business operator production lines" for clarity.

Importance

Line 51: Change "four CC, splitting each CC in" to "four CCs, splitting each CC into".

Introduction

Lines 60-65: Provide references for each statement.

Lines 82 and 83: Provide references.

Line 93: Change "the cost and the time" to "the cost and time".

Line 100: Change "rapidly detect of Lm" to "rapidly detect Lm".

Line 103: Change "(44-46), in" to "(44-46), and in".

Line 113: Change "multiplex PCR real-time systems" to "multiplex read-time PCR systems".

Line 115: Do "both assays" mean the high-throughput and conventional real-time PCR assays? Clarify this.

Materials and Methods

Line 123: Change "fluorimeter, DNA high" to "fluorimeter and a DNA high".

Line 124: Re-write "diluted, if necessary" for clarity.

Line 128: Throughout the manuscript, I recommend changing "ng.ul-1" to "ng/ul".

Line 132: Show the manufacturer of the lysozyme and its location.

Line 143: Change "k-mers position" to "k-mer positions".

Lines 144-145: Revise this since it is fragmentary.

Lines 148-152: Which program was used to download genomes from these databases? Remove "(EMBL-EBI)" in line 151, which is redundant with the acronym in line 152.

Line 153: Change "were produced" to "were sequenced".

Lines 157 and 158: Re-write this sentence, which does not make sense. Why is the quality of genomes discussed with genome sizes?

Line 158 and 159: What criteria were applied to select these representative genomes?

Lines 160 and 161: Re-write this statement to enhance clarity.

Line 162: Change "genomes assemblies" and "300 bp fragment (or k-mer)" to "genome assemblies" and "300 bp fragments (or k-mers)", respectively.

Lines 162 and 163: Change "specific & sensitive" to "specific and sensitive".

Line 163: Provide references for cdhit.

Line 164: I would recommend changing "retained" to "selected". Show criteria to select a single k-mer for each CC.

Lines 168 and 169: Remove the comma after "(the 4807 gene scheme)". I recommend incorporating the statement in lines 170 and 171 into here.

Lines 169 and 170: Re-write this statement for clarity. Also, which program or assay was used to select the alleles with the best real-time PCR discrimination?

Lines 174 and 175: I expected four numbers due to "between oligos, for hairpins, self-dimers and multiplex dimers" but only three were presented. Clarify this.

Line 182: Revise "and not only ... targeted CCs".

Lines 182 and 183: Which program was used to download fasta files of these genomes?

Lines 184 and 185: Change "6-Carboxyfluorescein" and "Hexachlorofluorescein" to "6-carboxyfluorescein" and "hexachlorofluorescein". Show what "ex." and "em." represent.

Line 201: "manually defined thresholds" sounds vague and I recommend showing specific thresholds and how they were determined.

Lines 205: Move "(57)" to the end of the statement (line 205).

Lines 207 and 208: It is hard to understand what "The plasmids ... recombinant plasmids" means. I suggest revising it to improve clarity.

Line 211-213: Revise this statement for clarity. How was genomic DNA concentration equivalence determined?

Lines 213-215: Revise this statement for clarity. Also, is "106 bp" a typo for "10⁶ bp"?

Line 215: Change "Each LoD tested" to "LoD test".

Line 217: Include "ISO 16140, part 4" to references and cite it here.

Line 218: Change "for the testing and validating the assay." to "to test and validate the assay".

Line 222: What is "-" for 587 "-"?

Line 228: Add a comma after "i.e."

Line 230: Remove "(2016)", which is not necessary.

Line 232: "remaining 117 strains" is confusing since I expected 418 strains (480 strains in total - 62 strains). Clarify this.
Line 233: I recommend merging the statement "They covered 43 STs (Table S1)" with the prior one.
Line 235: Provide references for "known to be frequent food-chain contaminants".
Line 243: Change "was based on" to "consisted of".
Line 247: Remove "(Premier Biosoft, San Francisco, USA)" since it was already shown in line 174.
Line 248: What does "similar" mean here? Any difference from the high-throughput real-time PCR assay regarding controls?
Line 251: Change "(Roche Diagnostics, Meylan, France) an" to "(Roche Diagnostics, Meylan, France) and".
Line 255: Include the manufacturer of master mix. Change "until" to "up to".
Line 258: It would be informative for readers if the authors mentioned how this threshold was determined.
Line 267: Change "Performance of the assay was performed" to "Performance of the assay was measured".
Line 274: Remove "Charbonnières-les-Bains, France", which was already shown in line 131.
Lines 276 and 277: Revise "a 2 min ... run time" to improve clarity.
Lines 281 and 282: What do all these acronyms (EURL, IT NRL and NL NRL) mean? Re-write this to improve clarity.

Results

Lines 290 and 291: Was this in silico analysis described in the Materials and Methods section? Which program was used?
Line 293: I would recommend changing "to be between" to ", ranging between".
Lines 295 and 296: What does this mean? Clarify and re-write this statement.
Line 304 and 320: What do "CC9 locus" and "CC203 locus" mean? Regions targeted in the real-time PCR to identify CC9 and Cc203? Also, it would be informative if the authors briefly described genes located in these loci.
Lines 309, 314 and 325: Clarify "both genomic panels".
Line 310: Change "was analyzed and generated" to "generated".
Line 313: Change "GuaA" to "guaA" and italicize it. Change other instances as well throughout the manuscript.
Lines 313 and 314: Why were these two strains mentioned here, not in the beginning of the paragraph along with strain 01CEB168LM?
Lines 314-316: Do "strains reported from Canada and France from fish products" mean two ST662 strains mentioned in lines 313 and 314? Revise this statement for clarity.
Line 316: Which strain was "strain genome" obtained from?
Line 319: I recommend moving "(Table S1)" to the end of the statement (line 320).
Line 321: Change "the internalin K gene" to "inlK" and italicize it.
Line 321: Clarify "deletion of the CC204 locus". Was "CC204 locus" replaced with the 43 kb genomic insertion? Or is the 43 kb genomic insertion located in the middle of the CC204 locus?
Lines 322 and 323: Re-write "ST204 strain deletion".
Line 324: What strains are referred to in "strain genomes"?
Line 333: I suggest changing "concorded" to "agreed".
Lines 334 and 335: This statement is vague and I recommend re-writing it.
Line 341: Clarify "remaining strains". Are they those whose CCs could not be determined by the high-throughput real-time PCR? Or are they those that were not isolated from food products, FPEs, or ruminants?
Lines 344-346: Cite tables or figures. Change "same plant" to "the same plant".
Line 350: I recommend "accounting for" to "including".
Lines 360-362: What does this statement mean? Re-write it to improve clarity.
Lines 365 and 371: Change "clonal complexes" to "CCs".
Lines 371-374: This part could be merged with lines 365-370. Change "all primer and probes sets" to "all primer and probe sets".
Line 377: Provide references.
Line 386: Change "less than 3 AD" to "strains with less than 3 AD".
Lines 388 and 389: This explanation sounds strange since the analysis output indicates that these processing plants were colonized by different strains of ST398.
Lines 390-392: Provide references.
Line 393: I believe that "audits" instead of "controls" might sound more appropriate here.
Line 399: Re-write "ST173 being part of CC20 PCR cross-reactions" to improve clarity.

Discussion

Line 409: Change "provides results" to "generates results".
Line 410: Add "and" before "(iv)".
Line 414: Only one in-house pipeline was mentioned in the Materials and Methods section. What are "two original bio-informatics tools"? Change "bio-informatics" to "bioinformatics".
Line 420: Clarify "following the molecular serotyping method developed by Doumith et al.". Do 28 primer and probe sets target genes utilized in the PCR serotyping assay devised by Doumith et al.? Also, remove "(2004)".
Lines 423 and 424: Change "the "subdivision" PCR" to "designed for "subdivision" PCR".
Lines 429 and 430: Change "respectively of ST9 (n=2) and ST204 (n=2)" to "of ST9 (n=2) and ST204 (n=2), respectively".
Lines 430 and 431: "Lmo1118" came from a locus tag of EGDe genome and was not named by Doumith et al. Revise it. Also, change "Lmo1118" to "lmo1118" throughout the manuscript.
Lines 431 and 432: I would recommend combining this statement with the one in lines 430 and 431.
Line 435: Lines 428-434 pertain to CC9 false-negative strains. Why are "CC9 false-positive strains" mentioned here?

Lines 435 and 436: Change "CC9 false-negatives" to "CC9 false-negative strains".
Lines 438 and 439: Clarify and re-write "on STs genetically ... CC193 strains". Minor STs belonging to CC193?
Line 439: What does "confirmed analytically" mean? Confirmed via the high-throughput real-time PCR? Change "a false-negative ST662" to "a false-negative ST662 strain".
Line 440: Does "observed in silico" mean that only in silico analysis was conducted or that it was found in silico but not confirmed with a wet lab assay?
Line 441: "with more than 1000 AD with the CC193 targeted strains" is confusing. Does this mean that ST662 and ST796 do not belong to CC193? Clarify and re-write this part. Also, I recommend starting a new paragraph from "All false-negative results".
Line 442: Remove the comma after "CC204" and change "the insertion or to the deletion" to "the insertion or deletion".
Lines 444 and 445: Revise this statement, which is fragmentary.
Lines 447 and 448: Change "resulting from our method" to "observed in our method".
Lines 452 and 453: Clarify and re-write "making molecular serotyping ... alternatives". What does "molecular serotyping" mean here? The PCR serotyping devised by Doumith et al.?
Lines 455-458: I suggest merging this paragraph with the prior one.
Lines 455 and 456: Clarify and re-write "confirmed analytically".
Line 456: Change "ST173 strain outbreak investigation" to "ST173 outbreak strain investigation".
Lines 459-466: I recommend moving this paragraph to after line 418.
Line 460: Change "international multi-country" to either "international" or "multi-country".
Line 461: Is "isolated" a typo for "isolates"?
Line 468: I suggest changing "tracking" to "trackdown of".
Line 469: Remove the colon after "consumables".
Lines 486 and 487: Change "the high-throughput and the conventional real-time PCR assay" to "the high-throughput and conventional real-time PCR assays".
Lines 491, 493, and 494: It will be helpful to readers if the authors add some explanations for the acronyms of the NRLs in different countries.
Lines 494 and 495: Revise this statement, which seems to pertain to the purpose of this study.
Line 497: Revise "in a timeframe ... mobile laboratory" to improve clarity.
Lines 502 and 503: Change "reaction 20 ul master mix volume" to "20 ul of reaction master mix".
Lines 504 and 505: Clarify and revise "pure strain DNA extracts" and "complex sample DNA extracts".
Line 521: Change "define" to "defining" and revise similar instances in lines 522-528.
Line 522: Revise "food Lm strain population structure" to "Lm population structure in foods".
Line 527: Change "and (v)" to "; and (v)".

Acknowledgments

Line 547: Change "respectively" to line 549 after "PCR analysis".

Funding

Lines 560, 563, and 566: Is "founded" a typo for "funded"?

Table 3

Line 905: Change "Listeria monocytogenes" to "L. monocytogenes".

Figure 1

Line 913: Change "flow chart" and "real time" to "flowchart" and "real-time", respectively.

Editor Comments:

In the article, "Identification by high-throughput real-time PCR of 30 major circulating *Listeria monocytogenes* clonal complexes in Europe," the authors describe the development of a high-throughput real-time PCR assay for the identification of 30 *Listeria monocytogenes* clonal complexes. They further adapt the assay to conventional PCR systems used in diagnostic labs. Overall, the significance and need for the study is well presented. Nevertheless, as this is an assay development study, additional explanations and details of the methods are required.

The assays were designed from and tested against several "panels" of *Listeria* strains and genomes, however, these different groups quickly became confusing and hard to distinguish (strains vs. genomes, # included, sources, used in which part of the study). Perhaps a table or labeling of the panels A, B, C, etc. with descriptions would aid the reader throughout the paper. Three different DNA extraction methods are described but the only result given was a statement in L337 that it did not affect the assay's performance. However, no information is given on the details of how this was determined. What was the starting amount of cells?

Probes were labeled with either FAM or HEX but no description is given to which dye was used for which probe or was one chosen over the other. Table 1 should show dye with probe sequence. Likewise, the conventional PCR assays were based on duplex and triplex reactions (L243) yet the groups of primers/probes used together in reactions are not described.

It doesn't appear the same set of strains or panel was analyzed by both the high-throughput assay as well as the 2 conventional assays. How then can the assays be compared between the 3 platforms?

Minor comments:

L32 - how can nomenclature provide info on persistence in food chain?

Overall sentence structure is awkward L28-57

All section headers need to be in same font

L96-97 rapid and reliable duplicated

L 100 remove "of"

L 108-116 no mention of Lm even though it's implied

L 144-145 incomplete sentence

L169 awkward "placed on the alleles"

L245 Cyanine was added to some probes. Again, which ones, how were they grouped?

L251 an should be "and"

Table 1: need dye used on probe sequence

Table 2: typo in header, "corss"

I didn't see Table S1 and S2.

Fig 1: I don't understand what this is supposed to show. The title says it's an interpretation of a multiplex PCR assay but the chart contents seem higher level with WGS and verification of concordance shown.

Staff Comments:

Preparing Revision Guidelines

Please return the manuscript within 60 days; if you cannot complete the modification within this time period, please contact me. If you do not wish to modify the manuscript and prefer to submit it to another journal, please notify me of your decision immediately so that the manuscript may be formally withdrawn from consideration by Microbiology Spectrum.

Reviewer comments:

Reviewer #1 (Comments for the Author):

Please see below the suggestions / comments.

Line-49: please put the hyphen between "real time" to make it uniform with other similar entries

-> Done

Lines-50-82-83: please put "s" to CC

-> Done

Line-121: please add more specifications on the 25 strains extracted

-> Specification were provided

Line-144: in the sentence missing the verb, please insert it

-> Done

Line-157: please check the correct wording of the sentence

-> Done

Lines 190-192: In these sentences you talk about 48 samples tested, while in the abstract and in line 110, you refer to 46 samples, check and possibly specify

-> The materials and methods section describes the type of chip used which is a 48:48 format. Details on the number of samples analysed per run are not needed and were removed.

Lines-230 and 436: standardize the way in which references are reported (year with or without brackets)

-> Done

Line-233: please put "s" to CC

-> Done

Line-251: please change "an" to "and"

-> Done

Line-356 (not line 352): please change 300 000 to 300,000

-> Done

Line-369: please insert the space between "the" and "77"

-> Done

Line-435: please put the hyphen between "false negatives" to make it uniform with other similar entries

-> Done

Line-441: please change 1000 to 1,000

-> Done

Line-465: please put "s" to CC

-> Done

Line-572 (not line 576) : please put "s" to strain

-> For "strain DNA" strain should not be plural

Line-604: please consider updating the reference (8) with the correct version referring to 2020 version

-> Date was updated in the text to 2019

Line-754: please consider updating the reference (51) with the correct version, Part 1 is referred only to Vocabulaire; moreover the link doesn't work

-> The references were changed for English version of the standards ISO16140 and ISO17025. Access to standard is charged and requires specific subscription. Therefore we preferred to removed the link to the French accreditation body website.

Line-896: please standardize the title character of Table 1

-> Done

Conclusions: please consider to add the limitation of the study or of the assays performed.

-> Done

Reviewer #2 (Comments for the Author):

This study describes two novel real-time PCR methods (high-throughput and multiplex PCRs), which can rapidly discriminate 30 clonal complexes that are frequently encountered in the food chain. The methods developed by the authors will definitely attract a great interest from the public health sector and food industry by allowing to identify the CC of a strain without relying on the whole genome sequencing. Also, primers and probes were systematically designed using a large panel of genomes and the novel methods were thoroughly validated in various aspects. However, the current manuscript could be further improved by addressing weaknesses shown below.

1. The conventional multiplex real-time PCR was not sufficiently described, at least not to the degree that a reader can replicate this method after reading this manuscript. For instance, "3 duplex and 10 triplex PCR reactions" in lines 243 and 244 were not specified in the text or tables and "several steps" comprising this real-time PCR (lines 249 and 250) were not adequately delineated. Also, Figure 1, which was cited in lines 249 and 250, did not clearly convey the process and thus needs to be revised. Given that this conventional real-time PCR method was developed so that diagnostic laboratories can perform this assay routinely, this could be a major pitfall that could reduce the usefulness of this study. In the Materials and Methods section in general, more details are needed. See specific comments.

-> The multiplex scheme was further detailed in table 1 with a multiplex code and dye and quencher indications. Figure 1 was simplified and made more explanatory for routine performance of the assay.

2. If the authors have included the features/functions of the inserted or deleted genes when discussing false-negative strains, it could have pleased many readers, including this reviewer, who are interested in the genetic diversity of *L. monocytogenes* population.

-> We have pushed the identification of integrative elements responsible of false negative identification for the PCR targeting CC9, CC193 and CC204. Unfortunately we did not find further annotation on them. When possible we identified the gene where the insertion or the deletion happened.

3. Show the GitHub site for the in-house bioinformatics pipeline, if available.

-> We provided the scientific article for BIOHIT tool, which is the core of our in-house pipeline. The rest of the analysis was performed in BioNumerics.

4. Many statements, in particular in the Results and Discussion sections, should be worked on to improve clarity. See specific comments.

-> Comments were followed to improve clarity.

5. References are often missing. See specific comments.

-> References were added.

6. Cite tables and figures more frequently in Results.

-> Done

Specific comments:

Keywords

Line 25: Change "Clonal complexes" to "Clonal Complexes".

-> Done

Abstract

Line 35: Change "serogroup of the strain" to "serogroup of a strain".

-> Done

Line 39: Change "its sensitivity and its specificity" to "its sensitivity and specificity".

-> Done

Lines 46-47 and 56-57: Re-write "food business operator production lines" for clarity.

-> Done

Importance

Line 51: Change "four CC, splitting each CC in" to "four CCs, splitting each CC into".

-> Done

Introduction

Lines 60-65: Provide references for each statement.

-> References were added

Lines 82 and 83: Provide references.

-> Reference were added

Line 93: Change "the cost and the time" to "the cost and time".

-> Done

Line 100: Change "rapidly detect of Lm" to "rapidly detect Lm".

-> Done

Line 103: Change "(44-46), in" to "(44-46), and in".

-> Done

Line 113: Change "multiplex PCR real-time systems" to "multiplex read-time PCR systems".

-> Done

Line 115: Do "both assays" mean the high-throughput and conventional real-time PCR assays? Clarify this.

-> Done

Materials and Methods

Line 123: Change "fluorimeter, DNA high" to "fluorimeter and a DNA high".

-> Done

Line 124: Re-write "diluted, if necessary" for clarity.

-> To clarify the sentence "diluted if necessary" was removed as dilution rules are specified in sections Results, 1.2 Limit of detection (LoD) and 2.1 Limit of detection.

Line 128: Throughout the manuscript, I recommend changing "ng.ul-1" to "ng/ul".

-> All units were modified accordingly

Line 132: Show the manufacturer of the lysozyme and its location.

-> Done

Line 143: Change "k-mers position" to "k-mer positions".

-> Done

Lines 144-145: Revise this since it is fragmentary.

-> Done

Lines 148-152: Which program was used to download genomes from these databases? Remove "(EMBL-EBI)" in line 151, which is redundant with the acronym in line 152.

-> The SRA were downloaded via the BioNumerics Calculation engine. Precision was added.

Line 153: Change "were produced" to "were sequenced".

-> Done

Lines 157 and 158: Re-write this sentence, which does not make sense. Why is the quality of genomes discussed with genome sizes?

-> *Lm* genomes size is between 2.8 and 3.1 Mb, assembly providing a genome out from this range must be related to a technical issue. The sentence was rephrased.

Line 158 and 159: What criteria were applied to select these representative genomes?

-> Selection criteria were added to the text.

Lines 160 and 161: Re-write this statement to enhance clarity.

-> The paragraph was modified to make it clearer.

Line 162: Change "genomes assemblies" and "300 bp fragment (or k-mer)" to "genome assemblies" and "300 bp fragments (or k-mers)", respectively.

-> Done

Lines 162 and 163: Change "specific & sensitive" to "specific and sensitive".

-> Done

Line 163: Provide references for cdhit.

-> Done

Line 164: I would recommend changing "retained" to "selected". Show criteria to select a single k-mer for each CC.

-> The selection of k-mer was done without pre-selection criteria. The k-mer selection was performed at step 2.1.3 using BLAST screening. The selection criteria were further specified in this section.

Lines 168 and 169: Remove the comma after "(the 4807 gene scheme)". I recommend incorporating the statement in lines 170 and 171 into here.

-> The paragraph was modified as well as section 2.1.3

Lines 169 and 170: Re-write this statement for clarity. Also, which program or assay was used to select the alleles with the best real-time PCR discrimination?

-> Specification were provided in particular software functionality used.

Lines 174 and 175: I expected four numbers due to "between oligos, for hairpins, self-dimers and multiplex dimers" but only three were presented. Clarify this.

-> Energy tolerance between oligos was not a nominal parameter. The sentence was modified for clarity.

Line 182: Revise "and not only ... targeted CCs".

-> The statement was revised.

Lines 182 and 183: Which program was used to download fasta files of these genomes?

-> Fasta files were directly uploaded from NCBI RefSeq assembly database, on the 28th of June 2019

Lines 184 and 185: Change "6-Carboxyfluorescein" and "Hexachlorofluorescein" to "6-carboxyfluorescein" and "hexachlorofluorescein". Show what "ex." and "em." represent. b

-> Excitation ("ex.") was replaced by absorbance. Modification was changed in the text.

Line 201: "manually defined thresholds" sounds vague and I recommend showing specific thresholds and how they were determined.

-> The manual threshold set up was specified in the text.

Lines 205: Move "(57)" to the end of the statement (line 205).

-> Done

Lines 207 and 208: It is hard to understand what "The plasmids ... recombinant plasmids" means. I suggest revising it to improve clarity.

-> The sentence was simplified, pBluescriptIIISK is the vector plasmid used to carry the PCR targeted sequenced.

Line 211-213: Revise this statement for clarity. How was genomic DNA concentration equivalence determined?

-> The information was added. The same methodology was applied for plasmid copy number estimation.

Lines 213-215: Revise this statement for clarity. Also, is "106 bp" a typo for "10⁶ bp"?

-> The calculation was false and replaced by the right formula.

Line 215: Change "Each LoD tested" to "LoD test".

-> Done

Line 217: Include "ISO 16140, part 4" to references and cite it here.

-> Done

Line 218: Change "for the testing and validating the assay." to "to test and validate the assay".

-> Done

Line 222: What is "- for 587 -"?

-> The hyphen were removed.

Line 228: Add a comma after "i.e.".

-> Done

Line 230: Remove "(2016)", which is not necessary.

-> Done

Line 232: "remaining 117 strains" is confusing since I expected 418 strains (480 strains in total - 62 strains). Clarify this.

-> The statement was modified to make it clearer.

Line 233: I recommend merging the statement "They covered 43 STs (Table S1)" with the prior one.

-> Done

Line 235: Provide references for "known to be frequent food-chain contaminants".

-> Reference were added

Line 243: Change "was based on" to "consisted of".

-> Done

Line 247: Remove "(Premier Biosoft, San Francisco, USA)" since it was already shown in line 174.

-> Done

Line 248: What does "similar" mean here? Any difference from the high-throughput real-time PCR assay regarding controls?

-> They are the ones used for HT real time PCR. The text was modified accordingly.

Line 251: Change "(Roche Diagnostics, Meylan, France) an" to "(Roche Diagnostics, Meylan, France) and".

-> Done

Line 255: Include the manufacturer of master mix. Change "until" to "up to".

-> Done

Line 258: It would be informative for readers if the authors mentioned how this threshold was determined.

-> Specifications were added.

Line 267: Change "Performance of the assay was performed" to "Performance of the assay was measured".

-> Done

Line 274: Remove "Charbonnières-les-Bains, France", which was already shown in line 131.

-> Done

Lines 276 and 277: Revise "a 2 min ... run time" to improve clarity.

-> The sentence was clarified. The TaqMan 7500 fast system global runtime was adjusted from 30 min to 55 min. 30 min was reported by mistake.

Lines 281 and 282: What do all these acronyms (EURL, IT NRL and NL NRL) mean? Re-write this to improve clarity.

-> The acronyms were developed.

Results

Lines 290 and 291: Was this in silico analysis described in the Materials and Methods section? Which program was used.

-> The in silico analysis strategy was further detail in Material and Methods section, cross detection identification was performed by BLAST as detailed in section 2.1.3.

Line 293: I would recommend changing "to be between" to ", ranging between".

-> Done

Lines 295 and 296: What does this mean? Clarify and re-write this statement.

-> Sentence was changed.

Line 304 and 320: What do "CC9 locus" and "CC203 locus" mean? Regions targeted in the real-time PCR to identify CC9 and Cc203? Also, it would be informative if the authors briefly described genes located in these loci.

-> The statements have been changed.

Lines 309, 314 and 325: Clarify "both genomic panels".

-> Done

Line 310: Change "was analyzed and generated" to "generated".

-> Done

Line 313: Change "GuaA" to "guaA" and italicize it. Change other instances as well throughout the manuscript.

-> Done

Lines 313 and 314: Why were these two strains mentioned here, not in the beginning of the paragraph along with strain 01CEB168LM?

-> Both ST662 strains were cited apart from 01CEB168LM because we only have the assemblies for these strains and we never tested them by PCR. The statement was modified to make it clearer.

Lines 314-316: Do "strains reported from Canada and France from fish products" mean two ST662 strains mentioned in lines 313 and 314? Revise this statement for clarity.

-> The statement was revised to clarify.

Line 316: Which strain was "strain genome" obtained from?

-> The statement was revised to clarify.

Line 319: I recommend moving "(Table S1)" to the end of the statement (line 320).

-> Done

Line 321: Change "the internalin K gene" to "inlK" and italicize it.

-> Done

Line 321: Clarify "deletion of the CC204 locus". Was "CC204 locus" replaced with the 43 kb genomic insertion? Or is the 43 kb genomic insertion located in the middle of the CC204 locus?

-> The genetic location of CC204 locus was clarified in the text.

Lines 322 and 323: Re-write "ST204 strain deletion".

-> This statement was removed and replaced by a clearer description.

Line 324: What strains are referred to in "strain genomes"?

-> Precision was provided.

Line 333: I suggest changing "concorded" to "agreed".

-> Done

Lines 334 and 335: This statement is vague and I recommend re-writing it.

-> The statement has been changed.

Line 341: Clarify "remaining strains". Are they those whose CCs could not be determined by the high-throughput real-time PCR? Or are they those that were not isolated from food products, FPEs, or ruminants?

-> Precision was provided

Lines 344-346: Cite tables or figures. Change "same plant" to "the same plant".

-> Done

Line 350: I recommend "accounting for" to "including".

-> Done

Lines 360-362: What does this statement mean? Re-write it to improve clarity.

-> Done

Lines 365 and 371: Change "clonal complexes" to "CCs".

-> Done

Lines 371-374: This part could be merged with lines 365-370. Change "all primer and probes sets" to "all primer and probe sets".

-> Both sections were merged

Line 377: Provide references.

-> A reference was provided

Line 386: Change "less than 3 AD" to "strains with less than 3 AD".

-> Done

Lines 388 and 389: This explanation sounds strange since the analysis output indicates that these processing plants were colonized by different strains of ST398.

-> The argue is not supported as you said. The statement was removed.

Lines 390-392: Provide references.

-> There is no publicly available reference for this outbreak investigation

Line 393: I believe that "audits" instead of "controls" might sound more appropriate here.

-> Done

Line 399: Re-write "ST173 being part of CC20 PCR cross-reactions" to improve clarity.

-> Done

Discussion

Line 409: Change "provides results" to "generates results".

-> Done

Line 410: Add "and" before "(iv)".

-> Done

Line 414: Only one in-house pipeline was mentioned in the Materials and Methods section. What are "two original bio-informatics tools"? Change "bio-informatics" to "bioinformatics".

-> the reference to both bioinformatics pipelines were added in the text.

Line 420: Clarify "following the molecular serotyping method developed by Doumith et al.". Do 28 primer and probe sets target genes utilized in the PCR serotyping assay devised by Doumith et al.? Also, remove "(2004)".

-> Compare the primer design approach of Doumith et al. with the approach applied in this study was here only to compare the approach, that in both study target unique sequences on *Lm* chromosome. The statement was modified to clarify it.

Lines 423 and 424: Change "the "subdivision" PCR" to "designed for "subdivision" PCR".

-> The statement was changed

Lines 429 and 430: Change "respectively of ST9 (n=2) and ST204 (n=2)" to "of ST9 (n=2) and ST204 (n=2), respectively"

-> Done

Lines 430 and 431: "Lmo1118" came from a locus tag of EGDe genome and was not named by Doumith et al. Revise it. Also, change "Lmo1118" to "lmo1118" throughout the manuscript.

-> The statement was changed accordingly.

Lines 431 and 432: I would recommend combining this statement with the one in lines 430 and 431.

-> Statement were combined

Line 435: Lines 428-434 pertain to CC9 false-negative strains. Why are "CC9 false-positive strains" mentioned here?

-> It was a mistake, correction is done

Lines 435 and 436: Change "CC9 false-negatives" to "CC9 false-negative strains".

-> Done

Lines 438 and 439: Clarify and re-write "on STs genetically ... CC193 strains". Minor STs belonging to CC193?

-> The statement was modified.

Line 439: What does "confirmed analytically" mean? Confirmed via the high-throughput real-time PCR? Change "a false-negative ST662" to "a false-negative ST662 strain".

-> Precision were provided.

Line 440: Does "observed in silico" mean that only in silico analysis was conducted or that it was found in silico but not confirmed with a wet lab assay?

-> Yes it mean that *in silico* analysis conducted without wet lab confirmation. We changed the statement accordingly.

Line 441: "with more than 1000 AD with the CC193 targeted strains" is confusing. Does this mean that ST662 and ST796 do not belong to CC193? Clarify and re-write this part. Also, I recommend starting a new paragraph from "All false-negative results".

-> Precision was provided on the genetic link between ST662 and ST796 and the other CC193 strains. ST796 in silico analysis were mentioned in the Results section 1.3 line . Mention to this ST in the results section

Line 442: Remove the comma after "CC204" and change "the insertion or to the deletion" to "the insertion or deletion".

-> Done

Lines 444 and 445: Revise this statement, which is fragmentary.

-> Done

Lines 447 and 448: Change "resulting from our method" to "observed in our method".

-> Done

Lines 452 and 453: Clarify and re-write "making molecular serotyping ... alternatives". What does "molecular serotyping" mean here? The PCR serotyping devised by Doumith et al.?

-> The statement was clarified accordingly.

Lines 455-458: I suggest merging this paragraph with the prior one.

-> Done

Lines 455 and 456: Clarify and re-write "confirmed analytically".

-> Done

Line 456: Change "ST173 strain outbreak investigation" to "ST173 outbreak strain investigation".

-> Done

Lines 459-466: I recommend moving this paragraph to after line 418.

The paragraph was moved.

Line 460: Change "international multi-country" to either "international" or "multi-country".

-> Done

Line 461: Is "isolated" a typo for "isolates"?

-> It is a typo, isolated was changed for isolates

Line 468: I suggest changing "tracking" to "trackdown of".

-> Done

Line 469: Remove the colon after "consumables".

-> Done

Lines 486 and 487: Change "the high-throughput and the conventional real-time PCR assay" to "the high-throughput and conventional real-time PCR assays".

-> Done

Lines 491, 493, and 494: It will be helpful to readers if the authors add some explanations for the acronyms of the NRLs in different countries.

-> The name of the institutes hosting the NRLs were specified when mention for the first time and after cited as NRL + two letters country acronym.

Lines 494 and 495: Revise this statement, which seems to pertain to the purpose of this study.

-> The statement was removed.

Line 497: Revise "in a timeframe ... mobile laboratory" to improve clarity.

-> Done

Lines 502 and 503: Change "reaction 20 ul master mix volume" to "20 ul of reaction master mix".

-> Done

Lines and 505: Clarify and revise "pure strain DNA extracts" and "complex sample DNA extracts".

-> The term were changed to define them more accurately.

Line 521: Change "define" to "defining" and revise similar instances in lines 522-528.

-> Done

Line 522: Revise "food Lm strain population structure" to "Lm population structure in foods".

-> Done

Line 527: Change "and (v)" to "; and (v)".

-> Done

Acknowledgments

Line 547: Change "respectively" to line 549 after "PCR analysis".

-> Done

Funding

Lines 560, 563, and 566: Is "founded" a typo for "funded"?

-> Done

Table 3

Line 905: Change "Listeria monocytogenes" to "L. monocytogenes".

-> Done for all tables and figures

Figure 1

Line 913: Change "flow chart" and "real time" to "flowchart" and "real-time", respectively.

-> Done

Editor Comments:

In the article, "Identification by high-throughput real-time PCR of 30 major circulating *Listeria monocytogenes* clonal complexes in Europe," the authors describe the development of a high-throughput real-time PCR assay for the identification of 30 *Listeria monocytogenes* clonal complexes. They further adapt the assay to conventional PCR systems used in diagnostic labs. Overall, the significance and need for the study is well presented. Nevertheless, as this is an assay development study, additional explanations and details of the methods are required.

The assays were designed from and tested against several "panels" of *Listeria* strains and genomes, however, these different groups quickly became confusing and hard to distinguish (strains vs. genomes, # included, sources, used in which part of the study). Perhaps a table or labeling of the panels A, B, C, etc. with descriptions would aid the reader throughout the paper.

Three different DNA extraction methods are described but the only result given was a statement in L337 that it did not affect the assay's performance. However, no information is given on the details of how this was determined. What was the starting amount of cells?

Probes were labeled with either FAM or HEX but no description is given to which dye was used for which probe or was one chosen over the other. Table 1 should show dye with probe sequence.

Likewise, the conventional PCR assays were based on duplex and triplex reactions (L243) yet the groups of primers/probes used together in reactions are not described.

It doesn't appear the same set of strains or panel was analyzed by both the high-throughput assay as well as the 2 conventional assays. How then can the assays be compared between the 3 platforms?

-> Details on the biological and genomic resources were better described and referenced in the manuscript using two distinct denominations for genomic panel: GP-A and B and for strain panel: SP-C, D and E. Details were provided on extraction method assessment, the cell amount used to carry out these methods were specified, the 25 strains used for extraction method comparison were flagged in Table S1 and the results were further detailed. Probes were annotated with dyes and quencher and multiplex groups of primer and probe sets were provided in Table 1. Annotation of strain panel SP-C, D was reported in Table 2, Table 3 and Table S1 providing a link between the text and the strains used for the high-throughput assay as well as conventional assays validation.

Minor comments:

L32 - how can nomenclature provide info on persistence in food chain?

-> Recurrent isolation of strains assigned to the same CC in the same area provide an information on their presumable persistence. The presumable persistence nuances the statement.

Overall sentence structure is awkward L28-57

-> The section was re-written

All section headers need to be in same font

-> Done

L96-97 rapid and reliable duplicated

-> Repetition was removed,

L 100 remove "of"

-> Done

L 108-116 no mention of Lm even though it's implied

-> Done

L 144-145 incomplete sentence

-> Done

L169 awkward "placed on the alleles"

-> Done

L245 Cyanine was added to some probes. Again, which ones, how were they grouped?

-> All dyes and multiplex groups were specified in Table 1 and reference was made in the text.

L251 an should be "and"

-> Done

Table 1: need dye used on probe sequence

-> Dyes were specified on the sequence

Table 2: typo in header, "corss"

-> Done

I didn't see Table S1 and S2.

-> These tables are very large and were provided in PDF as supplementary material.

Fig 1: I don't understand what this is supposed to show. The title says it's an interpretation of a multiplex PCR assay but the chart contents seem higher level with WGS and verification of concordance shown.

-> The Figure was revised to dedicate it to the multiplex PCR interpretation.

March 27, 2023

Mx. Benjamin Félix

Anses

ANSES, European Union Reference Laboratory for *Listeria monocytogenes*, Laboratory for Food Safety, Salmonella and Listeria Unit, University of Paris-Est, Maisons-Alfort, France

14 rue pierre et marie Curie

Maisons-alfort 94701

France

Re: Spectrum03954-22R1 (Identification by high-throughput real-time PCR of 30 major circulating *Listeria monocytogenes* clonal complexes in Europe)

Dear Mx. Benjamin Félix:

Thank you for submitting your manuscript to Microbiology Spectrum.

As you will see from the reviewer's comments, there are still a lot of grammatical errors that needs to be fixed. I will suggest that the authors thoroughly look over the manuscript again before resubmitting.

Link Not Available

Sincerely,

Adelumola Oladeinde

Journals Department
Reviewer comments:

Reviewer #2 (Comments for the Author):

Most of the comments and suggestions from this reviewer have been answered and the revised manuscript has significantly improved due to the extensive efforts to provide experimental details and increase clarity of the manuscript. However, further edits are still needed for the current manuscript to enhance overall wording, tables and figures as shown in specific comments.

Line numbers in the specific comments are based on the merged file without track changes and it would be helpful for this reviewer if the authors pinpoint in the rebuttal the line numbers where they modified for each comment.

Specific comments:

Abstract

Line 44: Change "PCR real-time" to "real-time PCR".

Line 47: I recommend removing the comma after "operators".

Importance

Lines 49 and 51: Show the full name for MLST and CCs, respectively.

Introduction

Line 86: Change "CCs nomenclature" to "CC nomenclature".

Lines 85 and 86: Change "CCs classification" to "CC classification".

Line 99: Change "Such method" to "Such a method".

Line 106: Change "species identification or the five". to "the identification of species or the five".

Line 111: Change "30 Lm major circulating CCs" to "30 major Lm CCs"

Line 113: Change "the CCs, the molecular serotype" to "the CCs and the molecular serotype".

Materials and methods

Line 124: Remove the comma in "strains, from".

Line 132: Change "agar non-selective medium cultured 24h" to "non-selective agar medium cultured for 24 h".

Lines 136 and 144: Change "Gram positive" to "Gram-positive" for consistency.

Lines 138 and 145: Change "broth overnight" to "grown overnight".

Line 147: Change "TritonX-100" to "Triton X-100".

Line 157: Change "two bacterial genome public databases" to "two public bacterial genome databases".

Line 161: Change "the 6 December 2018" to "the 6th of December 2018".

Line 166 and 167: Change "Only genomes, consistent with Lm genomic size, between 2.8 and 3.1 Mb," to "Only genomes between 2.8 and 3.1 Mb, consistent with Lm genomic size,".

Line 169: Remove the comma after "The 954 GP-A genomes".

Line 182: Change "Primer and probes parameters" to "Primer and probe parameters".

Lines 187, 340 and 591: Add a comma before "respectively".

Lines 195 and 196: Change "false negative cross detection" to "false-negative cross-detection". Revise "False positive cross detection" as well.

Lines 198 and 199: Change "FAM maximum absorbance 495 maximum emission 520 nm" to "FAM, maximum absorbance at 495 nm and maximum emission at 520 nm". Revise lines 199-200 and 265 accordingly.

Line 210: Change "2.7 µL of (0.1-1 ng/µL) diluted DNA" to "2.7 µL of diluted DNA (0.1-1 ng/µL)".

Line 220: Change "pBluescriptII SK" to "pBluescript II SK"

Line 223: Remove the parenthesis next to "France")."

Lines 244 and 320: Add a comma after "i.e.".

Line 247: Change "reference, under" to "reference under which"

Line 250: Change "117 remainings" to "other 117 strains".

Line 259: Change "in France and in Hungary" to "in France and Hungary".

Line 287: Remove the comma after "SP-D".

Line 296: Change "30 sec" to "for 30 sec".

Line 297: Change "55-minute" to "55 min".

Results

Line 308: Change "designed in this study" to "designed for this study".

Line 310: Change "CC121" to "and CC121".

Line 311: Change "within the CC" to "within each CC".

Line 329: Change "observed on 2.8%" to "observed in 2.8%".

Line 339: Change "Canada, France" and "USA" to "Canada and France" and "the USA", respectively.

Line 344: Change "real- time" to "real-time".

Line 345: Change "carried by" to "located in".

Lines 359 and 360: Merge this statement with the prior paragraph. Change "SP-C closely related strains" to "closely related strains in SP-C".

Line 365: Change "not sequenced strains" to "unsequenced strains".

Line 392: Change "until a concentration" to "up to a concentration".

Line 396: Change "TaqMan 7500 fast system" to "a TaqMan 7500 fast system".

Lines 401 and 402: Combine this statement with the prior paragraph. Change "(SP-D)" to "(SP-D)".

Line 413: Change "their genome compared" to "their genomes were compared".

Line 428: Add a comma after "Therefore".

Discussion

Line 434: Change "developed in the study proved able" to "developed for this study proved to be able".
Line 435: Change "MLST clonal complexes" to "CCs".
Line 438: Add a semi-colon before "and (iv)".
Line 445: Change "from meat products The 30 CCs set covers" to "from meat products. However, the 30 CC set covers".
Line 467: Change "primers and probes set" to "primer and probe set".
Line 468: Change "as characteristics of" to "as a characteristic of".
Lines 465 and 474: Does "five false-negative results" in line 465 mean "five false-negative strains"? Also, show the number of false-negative CC193 strains in line 474.
Line 480: Change "and was not" to "but not".
Line 482: Change "insertion or to the deletion" to "insertion or deletion".
Line 491: "conventional serotyping methods" could sound confusing since some readers could think that it refers to the traditional serological serotyping method. I recommend revising this to avoid confusion.
Line 494: I recommend starting a new statement from "27 of them".
Line 495: Change "the 22 others" to "and the others".
Line 499: Change "track down the sources" to "trackdown of the sources".
Line 503: Change "encountered by the food" to "encountered in the food".
Lines 504-508: This statement summarizes the accomplishments achieved by this paper and I cannot understand why a paper on Biomark is cited here. Unless there is a due reason, I would recommend not citing any paper for this statement.
Lines 513-516: Combine this paragraph to the prior one.
Line 514: Change "on line database" to "online database".
Line 515: Change "Nevertheless the 30 CCs identify are" to "Nevertheless, the 30 CCs identified are".
Line 521: Change "the high-throughput and the conventional" to "the high-throughput and conventional".
Lines 535 and 536: Change "CCs identification" to "CC identification".
Lines 541 and 542: Change "strain DNA already isolated and purified" to "genomic DNA purified from an isolated strain".
Lines 544 and 545: Provide references.
Line 549: Merge "Analyses are currently underway" with the previous sentence.
Line 549: Change "may be able to investigate" to "may be able to be used to investigate".

Conclusions

Line 558: Change "The method do not cover the described the" to "The methods do not cover the described".
Line 560: Change "contribute at" to "contribute to".
Line 566: Add a semi-colon before "(iv)". Remove the space after "investigations".
Lines 567 and 568: Change "to select" and "selecting".

Acknowledgements

Line 570: Change "The authors thanks" to "The authors thank".

Funding

Lines 605 and 610: Change "founded" to "funded".

Author Contribution

Line 624: Change "contributed to the project design, contributed to the writing" to "contributed to the project design and the writing".

Table 1

Lines 978 and 979: I believe that "WGS, whole-genome sequencing; CC, clonal complex" should be shown after the table as a legend.

My understanding is that probes are marked with "_P" and primers are signified with "_F" or "_R". I recommend including this information to the legend.

Show which dye was used for each Cy5-labeled probe in the high-throughput real-time PCR assay.

Table 2

In the "Analytical sensitivity" column, is "0,992" a typo for "0.992"? If so, revise all the numbers in this column appropriately.

In the "Cross-reaction with analytical confirmation" column, why is there a comma before brackets in several rows while there is not in other rows?

I recommend changing "clonal complex" to "CC" and "strain panel C" to "SP-C".

"(1) figures reported beside CC or ST report" could be re-written to "The number within parentheses beside CC or ST reports".

Table 3

Line 989: I expected two panels since 526 and 77 strains were mentioned in line 988. I recommend changing "strain panel D" to "SP-D".

Figure 1

Lines 992 and 993: Change "follow" to "followed". Provide the reference for Vittulo et al.

Line 993: Start a new sentence from "CC id". Change "abbreviation and means" to "abbreviation for".

In Figure 1, what do the numbers in rectangles under each molecular serotype mean? CCs with the numbers are sequentially identified according to the numbers? How were these numbers determined?

Staff Comments:

Preparing Revision Guidelines

Please return the manuscript within 60 days; if you cannot complete the modification within this time period, please contact me. If you do not wish to modify the manuscript and prefer to submit it to another journal, please notify me of your decision immediately so that the manuscript may be formally withdrawn from consideration by Microbiology Spectrum.

Reviewer comments:

Reviewer #2 (Comments for the Author):

Most of the comments and suggestions from this reviewer have been answered and the revised manuscript has significantly improved due to the extensive efforts to provide experimental details and increase clarity of the manuscript. However, further edits are still needed for the current manuscript to enhance overall wording, tables and figures as shown in specific comments. Line numbers in the specific comments are based on the merged file without track changes and it would be helpful for this reviewer if the authors pinpoint in the rebuttal the line numbers where they modified for each comment.

Specific comments:

Abstract

Line 44: Change "PCR real-time" to "real-time PCR".

- Line 42-43

Line 47: I recommend removing the comma after "operators".

- Line 46

Importance

Lines 49 and 51: Show the full name for MLST and CCs, respectively.

- Line 48 and 50, the section was modified to comply word count.

Introduction

Line 86: Change "CCs nomenclature" to "CC nomenclature".

- Line 82

Lines 85 and 86: Change "CCs classification" to "CC classification".

- Line 84

Line 99: Change "Such method" to "Such a method".

- Line 97

Line 106: Change "species identification or the five". to "the identification of species or the five".

- Line 104

Line 111: Change "30 Lm major circulating CCs" to "30 major Lm CCs"

- Line 109

Line 113: Change "the CCs, the molecular serotype" to "the CCs and the molecular serotype".

- Line 111

Materials and methods

Line 124: Remove the comma in "strains, from".

- Line 122

Line 132: Change "agar non-selective medium cultured 24h" to "non-selective agar medium cultured for 24 h".

- Line 130

Lines 136 and 144: Change "Gram positive" to "Gram-positive" for consistency.

- Line 134, line 142

Lines 138 and 145: Change "broth overnight" to "grown overnight".

- Line 136

Line 147: Change "TritonX-100" to "Triton X-100".

- Line 145

Line 157: Change "two bacterial genome public databases" to "two public bacterial genome databases".

- Line 155

Line 161: Change "the 6 December 2018" to "the 6th of December 2018".

- Line 159

Line 166 and 167: Change "Only genomes, consistent with Lm genomic size, between 2.8 and 3.1 Mb," to "Only genomes between 2.8 and 3.1 Mb, consistent with Lm genomic size,".

- Line 164-165

Line 169: Remove the comma after "The 954 GP-A genomes".

- Line 168

Line 182: Change "Primer and probes parameters" to "Primer and probe parameters".

- Line 181

Lines 187, 340 and 591: Add a comma before "respectively".

- Line 185, 332 and 579

Lines 195 and 196: Change "false negative cross detection" to "false-negative cross-detection".
Revise "False positive cross detection" as well.

- Line 192-193

Lines 198 and 199: Change "FAM maximum absorbance 495 maximum emission 520 nm" to "FAM, maximum absorbance at 495 nm and maximum emission at 520 nm". Revise lines 199-200 and 265 accordingly.

- Line 196 - 198

Line 210: Change "2.7 µL of (0.1-1 ng/µL) diluted DNA" to "2.7 µL of diluted DNA (0.1-1 ng/µL)".

- Line 207

Line 220: Change "pBluescriptIISK" to "pBluescript II SK"

- Line 217

Line 223: Remove the parenthesis next to "France)."

- Line 219

Lines 244 and 320: Add a comma after "i.e."

- Line 238 and 312

Line 247: Change "reference, under" to "reference under which"

- Line 241

Line 250: Change "117 remainings" to "other 117 strains".

- Line 244

Line 259: Change "in France and in Hungary" to "in France and Hungary".

- Line 253

Line 287: Remove the comma after "SP-D".

- Line 280

Line 296: Change "30 sec" to "for 30 sec".

- Line 289

Line 297: Change "55-minute" to "55 min".

- Line 290

Results

Line 308: Change "designed in this study" to "designed for this study".

- Line 301

Line 310: Change "CC121" to "and CC121".

- Line 303

Line 311: Change "within the CC" to "within each CC".

- Line 304

Line 329: Change "observed on 2.8%" to "observed in 2.8%".

- Line 322

Line 339: Change "Canada, France" and "USA" to "Canada and France" and "the USA", respectively.

- Line 332

Line 344: Change "real- time" to "real-time".

- Line 337

Line 345: Change "carried by" to "located in".

- Line 338

Lines 359 and 360: Merge this statement with the prior paragraph. Change "SP-C closely related strains" to "closely related strains in SP-C".

- Line 351-352

Line 365: Change "not sequenced strains" to "unsequenced strains".

- Line 358

Line 392: Change "until a concentration" to "up to a concentration".

- Line 384

Line 396: Change "TaqMan 7500 fast system" to "a TaqMan 7500 fast system".

- Line 393

Lines 401 and 402: Combine this statement with the prior paragraph. Change "(SP-D)" to "(SP-D)".

- Line 392-393

Line 413: Change "their genome compared" to "their genomes were compared".

- Line 405

Line 428: Add a comma after "Therefore".

- Line 419

Discussion

Line 434: Change "developed in the study proved able" to "developed for this study proved to be able".

- Line 425

Line 435: Change "MLST clonal complexes" to "CCs".

- Line 426

Line 438: Add a semi-colon before "and (iv)".

- Line 429

Line 445: Change "from meat products The 30 CCs set covers" to "from meat products. However, the 30 CC set covers".

- Line 436

Line 467: Change "primers and probes set" to "primer and probe set".

- Line 457

Line 468: Change "as characteristics of" to "as a characteristic of".

- Line 458

Lines 465 and 474: Does "five false-negative results" in line 465 mean "five false-negative strains"? Also, show the number of false-negative CC193 strains in line 474.

- Line 455-456 and 464-465, the statements were modified.

Line 480: Change "and was not" to "but not".

- Line 470

Line 482: Change "insertion or to the deletion" to "insertion or deletion".

- Line 473

Line 491: "conventional serotyping methods" could sound confusing since some readers could think that it refers to the traditional serological serotyping method. I recommend revising this to avoid confusion.

- Line 482 conventional was changed to molecular.

Line 494: I recommend starting a new statement from "27 of them".

- Line 485-486

Line 495: Change "the 22 others" to "and the others".

- Line 486

Line 499: Change "track down the sources" to "trackdown of the sources".

- Line 490

Line 503: Change "encountered by the food" to "encountered in the food".

- Line 494

Lines 504-508: This statement summarizes the accomplishments achieved by this paper and I cannot understand why a paper on Biomark is cited here. Unless there is a due reason, I would recommend not citing any paper for this statement.

- The reference was removed

Lines 513-516: Combine this paragraph to the prior one.

- Line 503

Line 514: Change "on line database" to "online database".

- Line 505

Line 515: Change "Nevertheless the 30 CCs identify are" to "Nevertheless, the 30 CCs identified are".

- Line 505-506

Line 521: Change "the high-throughput and the conventional" to "the high-throughput and conventional".

- Line 512

Lines 535 and 536: Change "CCs identification" to "CC identification".

- Line 526

Lines 541 and 542: Change "strain DNA already isolated and purified" to "genomic DNA purified from an isolated strain".

- Line 531-532

Lines 544 and 545: Provide references.

- Two references were provided

Line 549: Merge "Analyses are currently underway" with the previous sentence.

- Line 539

Line 549: Change "may be able to investigate" to "may be able to be used to investigate"

- Line 540

Conclusions

Line 558: Change "The method do not cover the described the" to "The methods do not cover the described".

- Line 549

Line 560: Change "contribute at" to "contribute to".

- Line 551

Line 566: Add a semi-colon before "(iv)". Remove the space after "investigations".

- Line 557

Lines 567 and 568: Change "to select" and "selecting".

- Line 559

Acknowledgements

Line 570: Change "The authors thanks" to "The authors thank".

- Line 561

Funding

Lines 605 and 610: Change "founded" to "funded".

- Line 596 to 601

Author Contribution

Line 624: Change "contributed to the project design, contributed to the writing" to "contributed to the project design and the writing".

- Line 615

Table 1

Lines 978 and 979: I believe that "WGS, whole-genome sequencing; CC, clonal complex" should be shown after the table as a legend.

CC was added in column header, WGS was removed as not cited in the table 1.

My understanding is that probes are marked with "_P" and primers are signified with "_F" or "_R". I recommend including this information to the legend.

- Done, in foot note "c"

Show which dye was used for each Cy5-labeled probe in the high-throughput real-time PCR assay.

- Alternative dyes used for the high throughput real-time PCR assay were reported under bracket in Table 1 following footnote "b".

Table 2

In the "Analytical sensitivity" column, is "0,992" a typo for "0.992"? If so, revise all the numbers in this column appropriately.

- Done

In the "Cross-reaction with analytical confirmation" column, why is there a comma before brackets in several rows while there is not in other rows?

- It was a typo, comma were removed

I recommend changing "clonal complex" to "CC" and "strain panel C" to "SP-C".

- Done

"(1) figures reported beside CC or ST report" could be re-written to "The number within parentheses beside CC or ST reports".

- Done

Table 3

Line 989: I expected two panels since 526 and 77 strains were mentioned in line 988. I recommend changing "strain panel D" to "SP-D".

- Line 250, 281, 359, 393 and 981-982: the legend was modified by defining two SP-D subpanels, SP-D.1 including 526 strains and SP-D.2 including 77 strains. Both were used for performance assessment of the high throughput and the conventional multiplex real-time PCR assay, respectively.

Figure 1

Lines 992 and 993: Change "follow" to "followed". Provide the reference for Vittulo et al.

- Line 986, the reference was provided

Line 993: Start a new sentence from "CC id". Change "abbreviation and means" to "abbreviation for".

- Line 987

In Figure 1, what do the numbers in rectangles under each molecular serotype mean? CCs with the numbers are sequentially identified according to the numbers? How were these numbers determined?

- The number in each square left corner refers to the multiplex number reported in Table 1. This is a linear numbering. The legend was completed accordingly.

April 14, 2023

Mx. Benjamin Félix

Anses

ANSES, European Union Reference Laboratory for *Listeria monocytogenes*, Laboratory for Food Safety, Salmonella and Listeria Unit, University of Paris-Est, Maisons-Alfort, France

14 rue pierre et marie Curie

Maisons-alfort 94701

France

Re: Spectrum03954-22R2 (Identification by high-throughput real-time PCR of 30 major circulating *Listeria monocytogenes* clonal complexes in Europe)

Dear Mx. Benjamin Félix:

Thank you for the opportunity to review your manuscript and I applaud the effort put into this study and your patience during the review process.

Your manuscript has been accepted, and I am forwarding it to the ASM Journals Department for publication. You will be notified when your proofs are ready to be viewed.

Sincerely,

Adelumola Oladeinde
